# Single-cell dynamics of genome-nucleolus interactions captured by nucleolar laser microdissection (NoLMseq)

Kaivalya Walavalkar ®[1], Shivani Gupta[1], Jelena Kresoja-Rakic[1], Mathieu Raingeval[1,2], Chiara Mungo[1,2], Philip Rubin[1,2] & Raffaella Santoro ®[1] ✉

Gene positioning in nuclear space is central to regulation, with repressive chromatin domains often contacting the nuclear lamina or nucleoli. The nucleolus undergoes structural changes in different cellular states, potentially altering genome organization. Yet, how nucleolar states influence 3D-genome architecture remains underexplored, largely due to the lack of methods able to map nucleolar-associated domains (NADs) in single cells and stressed nucleoli. We developed NoLMseq, a technique combining laser-capture microdissection and DNA sequencing to map NADs in single cells. NoLMseq uncovered unexplored features of chromosome organization around nucleoli, including NAD heterogeneity in mouse embryonic stem cells, yielding two major populations with distinct chromatin and developmental states. NADs predominantly contact nucleoli monoallelically, with contact frequency correlating with gene expression and chromatin states. Under nucleolar stress, NoLMseq revealed extensive chromosome reorganization, highlighting the importance of nucleolus integrity in genome organization. Thus, NoLMseq provides a critical tool to study 3D-genome responses to nucleolar states in health and disease.

The three-dimensional (3D) organization of chromosomes in the cell's nucleus plays an important role in the regulation of gene expression programs and cell fate. One important aspect of this regulation is the position of genes in the nuclear space. This is exemplified by the frequent location of repressive chromatin domains at the nuclear periphery (i.e., lamina-associated domains, LADs) or around nucleoli (nucleolar-associated domains, NADs)[1–3]. However, while LADs have been extensively studied in many cell types and in single cells, NADs have remained underinvestigated mainly due to technical limitations in identifying DNA sequences associated with a compartment lacking a membrane, such as the nucleolus.

The nucleolus exhibits a highly dynamic structure that changes in response to external stimuli affecting ribosome biogenesis, a critical nucleolar process that is tightly regulated according to cell state[4]. These nucleolar structural alterations (i.e., size, number, or physico-chemical properties) might significantly impact the organization of the surrounding genome, thereby affecting gene expression and chromatin states. Ribosome biogenesis is initiated in the nucleolus by the RNA polymerase I (Pol I)-driven transcription of hundreds of ribosomal RNA (rRNA) genes (rDNA) that generate 47S pre-rRNA. These transcripts are then modified, processed, and assembled with ribosomal proteins (RPs) in the nucleolus, forming pre-ribosomal particles[5]. rRNA genes are distributed among different chromosomes; acrocentric chromosomes in human cells and, in mouse cells, depending on strain, they can generally be found at chromosomes 12, 16, 18, and 19 (rDNA-chromosomes) and close to centromeric regions[3]. Due to ribosome biogenesis, the nucleolus structure is influenced by cell states such that highly proliferative cells have larger nucleoli compared to quiescent cells[6–9]. Accordingly, alterations in nucleolar activities and structure have been well documented under many cellular stress conditions

[1]Department of Molecular Mechanisms of Disease, DMMD, University of Zurich, Zurich CH-8057, Switzerland. [2]Molecular Life Science Program, Life Science Zurich Graduate School, University of Zurich, Zurich CH-8057, Switzerland. ✉e-mail: raffaella.santoro@dmmd.uzh.ch

and in several diseases, such as cancer, neurodegenerative disorders, and premature ageing[10]. For example, nucleolar stress is defined as a condition in which abnormalities in nucleolar structure and function driven by diverse cellular insults, such as nutrient starvation or DNA damage, ultimately lead to activation of stress signaling pathways that downregulate ribosome biogenesis, including rRNA gene transcription, and alter nucleolar structure[11,12]. However, whether these structural alterations might have an impact on the genome organization around nucleoli remains unexplored, mainly due to the technical limitations of current methods to map NADs.

Currently, there are several methods to identify NADs: the biochemical purification of nucleoli followed by sequencing and, more recently, the Nucleolar-DamID that relies on the expression of an engineered nucleolar histone H2B fused to Dam that serves to mark NADs with adenine methylation (H2B-Dam-NoLS), nucleolar-TSAseq, and O-MAP[13-15]. Although based on different methodologies, these methods revealed that NADs are composed of repressive chromatin domains[16-20]. However, these methods also have some limitations. First, they cannot be applied to study how genome organization around nucleoli is affected under nucleolar stress[21,22]. Indeed, under these conditions, nucleoli become frail and cannot be efficiently biochemically purified. On the other side, the consequent downregulation of ribosome biogenesis decreases protein translation, thereby affecting the expression of H2B-Dam-NoLS. Moreover, the physico-chemical alterations of nucleoli upon rDNA downregulation cause the expulsion of the H2B-Dam-NoLS from nucleoli, similarly to other nucleolar components[20]. Finally, these methods have only been applied in cell populations, and consequently, they have provided an average nucleolar contact frequency, which might not accurately reflect the structure of the genome around single nucleoli. Thus, so far, there is no methodology able to identify NADs under nucleolar stress conditions and in single cells.

To overcome these challenges, we developed NoLMseq, a novel approach that combines laser-capture microdissection (LCM) and DNA sequencing to identify NADs in single cells. LCM is a technique traditionally used for isolating single cells or specific tissue regions[23-27], and we have expanded it for the organelle-level isolation of single nucleoli. Traditional techniques to study genome organization predominantly rely on either microscopy or DNA sequencing. NoLMseq synergistically integrates the strengths of both these approaches, enabling precise examination of the genome architecture surrounding single nucleoli. The application of NoLMseq in mouse embryonic stem E14 cells (ESCs) not only confirmed the general repressive state of NADs but also provided novel insights into genome organization around nucleoli that could not be detected in previous NAD studies, since these were based on methods that could only be applied in cell populations. These findings included the detection of different levels of NAD heterogeneity among ESCs, resulting in two major populations with distinct chromatin and developmental states. NoLMseq also determined that genes detaching from nucleoli during ESC differentiation into neural progenitors are activated and linked to neuronal pathways, suggesting an active role of the nucleolus in gene repression. Intriguingly, although NADs are generally composed of repressive chromatin domains, NoLMseq detected a high frequency association of the highly expressed ribosomal protein genes with the nucleolus. These findings indicate that the repressive nucleolar compartment can also accommodate transcriptionally active genes and suggest a potential role of nucleolar proximity in regulating ribosomal protein genes and, consequently, ribosome biogenesis. Moreover, NoLMseq revealed that NADs prevalently contact nucleoli in a monoallelic manner and that allelic nucleolar contact frequency correlates with gene expression and chromatin states. Finally, we determined how genome structure is impacted under conditions that impair nucleolar integrity and that the loss of nucleolar integrity might play a significant role in the derepression of NAD-genes. This insight, unattainable with previous technologies, underscores the critical role of nucleolar integrity in genome organization and gene expression, while also illuminating how the genome responds to nucleolar stress.

Together, these results not only provided novel insights into how chromosomes are organized around nucleoli and their remodeling during cell fate commitment and stress-induced nucleolar structural changes but also demonstrated that NoLMseq accurately measures chromosome contacts around single healthy and "stressed" nucleoli. Thus, NoLMseq will be a critical tool to study NADs and determine how genome organization responds to nucleolar stress in healthy and disease states.

## Results

### NoLMseq identifies NADs in single cells

To map NADs at a single-nucleoli resolution, we combined laser-capture microdissection (LCM) of nucleoli and DNA sequencing. We named this method nucleolus laser microdissection sequencing (NoLMseq). We applied NoLMseq to ESCs, enabling comparisons with NADs recently identified in ESC populations using Nucleolar-DamID[20]. Moreover, the majority of ESCs display a single nucleolus[28], allowing for only one microdissection per cell. ESCs were grown on a thin polyester membrane that can be cut with a fine laser (Fig. 1a, b, Supplementary Fig. 1a). In brightfield microscopy, the nucleolus appears as a distinct, dark, and dense region within the nucleus, allowing its visualization and excision using laser capture microdissection (LCM). The isolated nucleolus was then collected into the cap of a microcentrifuge tube using MMI CapSure technology. The DNA content of each nucleolus was purified and amplified via quasilinear whole-genome amplification (WGA)[29]. To prevent any loss of DNA, WGA was carried out in the same tube in which the microdissected nucleolus was collected. Before proceeding with library preparation and DNA sequencing, the purity of nucleolar DNA was assessed by PCR to determine the presence of rRNA genes, which are located within nucleoli, and the absence of *Tuba1*, an active gene that previous studies have shown not to contact nucleoli[20] (Supplementary Fig. 1b).

We generated 53 high-quality single-nucleolus contact maps from ESCs, with a median of $5 \times 10^5$ uniquely mapped reads per nucleolus (Fig. 1c, Supplementary Figs. 1c and 2). We binned the genome into 100 kb segments and called NADs using SICER2[30] at a 100 kb resolution using genomic DNA as a control. The identified reads over the reference genome indicate that loci are either in a "contact" or "no contact" state with nucleoli (Fig. 1c, Supplementary Fig. 1c). Imaris analysis of 3D immunofluorescence images revealed that approximately 65% of the volume within the microdissected cylinder is occupied by the nucleolus, marked by the nucleolar protein NPM1, and the surrounding NADs, which were recently shown to form a -0.5–0.6 μm-wide ring around the nucleolus marked by H3K9me2[31] (Supplementary Fig. 1d). To exclude the inevitable presence of genomic sequences above and below nucleoli and NADs, we analyzed the data by calculating the nucleolar contact frequency (CF) that we defined as the proportion of microdissected nucleoli containing a defined NAD sequence (Fig. 1d). For all downstream analyses, we selected NADs with CF > 20%. By applying this cut-off in CF to define NADs and considering the nucleolus and NAD volume in each microdissected cell, the probability that a non-NAD region can be falsely classified as a NAD is only 0.000144% (FDR $1.44 \times 10^{-6}$) (see Method).

We found that NADs from rDNA-chromosomes have higher CF than chromosomes not containing rRNA genes, underscoring the specificity of NoLMseq in identifying NADs. Moreover, chromosome X, which is active in the analyzed male ESCs, showed the lowest nucleolar CF. This pattern could not be obtained using 53 random samples with genome coverage matching the genome coverage of each of the 53 ESC nucleoli measured by NoLMseq, supporting the specificity of our analysis (Supplementary Figs. 1e and 2).

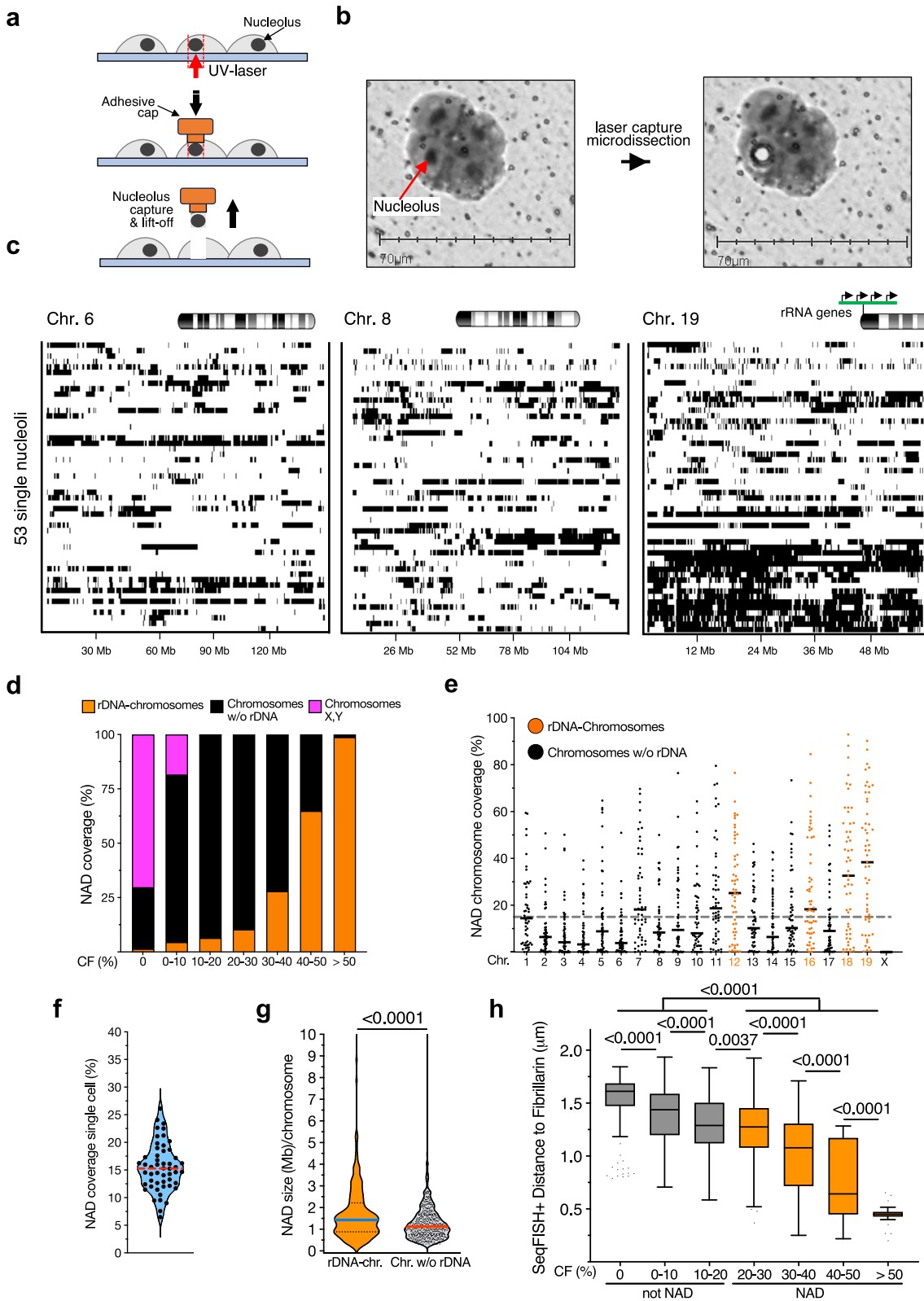

The analysis of NAD coverage/chromosome of all 53 nucleoli further supported the high propensity of rDNA-chromosomes to contact nucleoli relative to the other chromosomes (Fig. 1e). Furthermore, the analysis of 53 microdissected nucleoli appears to be sufficient for robust NAD identification. Calculation of nucleolar contact frequencies and genome coverage using subsets of 20, 30, and 40 randomly selected nucleoli indicates that NAD saturation is reached by 40 nucleoli (Supplementary Fig. 1f, g).

The average NAD coverage in single nucleoli was about 15%, much lower than the 30% NAD coverage obtained from previous NAD studies in ESC population[18,20] (Fig. 1e, f). The data also showed that the average length of NADs of rDNA-chromosomes was 1.85 Mb and significantly

**Fig. 1 | NoLMseq identifies NADs in single cells. a** Scheme representing the nucleolar laser capture microdissection (NoLMseq). **b** Representative images of ESCs before and after nucleolar laser capture microdissection. **c** Nucleolar contacts maps from 53 single nucleoli for chromosomes 6, 8, and 19. Black bars indicate NADs, white bar is for a non-NAD region. **d** Cumulative histogram of genome-wide nucleolar contact frequency (CF) values with respect to their location at chromosomes containing rRNA genes (rDNA-chromosomes), without (w/o rDNA), and X and Y chromosomes. **e** Coverage of NADs with CF > 20% on each chromosome in single nucleoli. The dotted gray line indicates the mean NAD coverage in single cells. Black lines indicate the mean on each chromosome. **f** Violin plot showing NAD coverage in single cells. The red dashed line indicates the median. **g** Size of NADs

located at rDNA-chromosomes and chromosomes not bearing rRNA genes (w/o rDNA). Red and blue lines indicate medians, and the dotted lines indicate quartiles. Statistical significance (P-values) was calculated using a two-sided unpaired t-test. **h** Genome-wide comparison of DNA seqFISH+ distance to exterior of nucleoli marked with Fibrillarin (μm)[32] and NoLMseq NAD contact frequency for 3713 paired genomic regions. Tukey boxplot where box limits represent the 25th and 75th percentiles and the black line represents the median. Statistical significance (P-values) was calculated using a two-sided unpaired t-test. Box plots depict the minimum and maximum values. The horizontal line within the boxes represents the mean value. Source data are provided as a Source Data file.

higher than the average NAD size from the rest of the chromosomes (1.30 Mb, Fig. 1g). To validate the data, we used an orthogonal approach by computing the CF of NADs from NoLMseq genomic data and their distance to Fibrillarin marked nucleoli obtained by seqFISH+ microscopy[32] (Fig. 1h). We observed that nucleolar CFs measured by NoLMseq were inversely proportional to the distance of NADs to the nucleolar marker Fibrillarin, indicating that NoLMseq accurately measures genomic distance to nucleoli.

To further support the efficacy of NoLMseq, we analyzed known NAD features of the identified NADs. It is well known that centromeres tend to be positioned close to the nucleolus in various model systems. Since in mouse cells all chromosomes are acrocentric, we measured the average NAD coverage at centromere-proximal regions at the 5′-quintile portion of all chromosomes and found that it increases with nucleolar CF, further supporting the accuracy of NoLMseq to identify sequences in contact with nucleoli (Fig. 2a). Consistent with previous results in ESC populations obtained with Nucleolar-DamID[20], NADs identified by NoLMseq were significantly depleted of active histone marks relative to sequences within the active A compartment, showed a significant enrichment for the repressive H3K9me2 but not for the other repressive marks H3K27me3 and H3K9me3, and displayed low gene expression levels (Fig. 2b, c). Although NADs generally appear to have a repressive chromatin and gene expression state, we could also find cases where highly expressed genes have high nucleolar CFs. In particular, we found high nucleolar CFs for ribosomal protein (RP) genes, which encode proteins for ribosome assembly in nucleoli and are transcribed by RNA Pol II, and tRNA genes, which are transcribed by Pol III (Fig. 2d). In contrast, snoRNA genes, which express RNAs playing an important role in rRNA modifications in nucleoli, did not show any preferential location to nucleoli. The nucleolar contacts of RP and tRNA genes were not due to their location at rDNA-chromosomes since most of them were located at chromosomes non containing rRNA genes (92% and 90%, respectively; Supplementary Fig. 3a). Consistent with this result, we reanalyzed previous Nucleolar-DamID data in ESC population[20] and found that 31% of RPs could be detected at NADs. Finally, the CFs of ribosomal protein genes with nucleoli from seqFISH+ data[32] highly correlate with the corresponding NoLMseq CF values (Supplementary Fig. 3b). These results raise the possibility that nucleolus proximity could play a role in regulating RP genes and ribosome biogenesis, as well as tRNA, potentially influencing protein translation on a broader scale. Finally, these data also suggested that not all NADs are necessarily transcriptionally repressed.

NADs have been previously characterized in two categories: NAD-only (or Type-1), which correspond to sequences contacting nucleoli but not the nuclear lamina (NL) and NAD/LAD (or Type-2), which are regions that can be located both at nucleoli and NL[13,18–20]. The analysis of single nucleoli data revealed that NAD-only sequences have higher CF values than NAD/LAD (Fig. 2e). This is consistent with the fact that NAD-only regions are not associated with the NL and are therefore more frequently localized to nucleoli than NAD/LAD regions, which can also associate with the NL. This result underscores the accuracy of NoLMseq in identifying NADs. Moreover, the data suggest that NAD/

LAD domains can contact the nucleolus in one cell and the NL in another cell. We also observed a small but statistically significant inverse correlation between NAD-only contact frequency (CF) and H3K27ac levels, particularly for regions with CF > 50%, indicating that strong NADs tend to exhibit low levels of active chromatin (Fig. 2f, Supplementary Fig. 3c). Similarly, NAD-only domains with high nucleolar CF have high H3K9me2 levels (Fig. 2g, Supplementary Fig. 3d). However, while NAD/LAD regions showed significantly lower gene expression levels relative to genes within the active A compartment, the expression of genes within NAD-only regions was not significantly reduced, albeit there was a tendency to be low expressed when located at regions with the highest nucleolar CF (Fig. 2h, Supplementary Fig. 3e). These results are consistent with the data described above showing that some active genes can also contact nucleoli.

## Detachment of NADs upon differentiation into NPCs correlates with gene activation

To determine whether NoLMseq could be applied to cells other than ESCs, we microdissected nucleoli of neural progenitors (NPCs) derived from the differentiation of ESCs, also taking advantage of recent Nucleolar-DamID analyses of NPCs for data validation[20]. As in the case of ESCs, in NPCs, NADs from rDNA chromosomes consistently exhibited higher nucleolar CF than regions from other chromosomes (Fig. 3a). However, NPCs displayed fewer NADs than ESCs, with a large proportion (75%) being the same NADs as found in ESCs (Fig. 3b–d). To assess whether the lower NAD coverage in NPCs is a result of increased noise, we performed an autocorrelation analysis on single nucleoli from ESCs and NPCs. The analysis revealed that autocorrelation decay patterns of ESCs (initial autocorrelation $a = 0.74$, decay constant $b = 0.23$) and NPCs ($a = 0.64$, $b = 0.34$) substantially differ from values generated with random sequences for ESCs and NPCs. These results indicated that the observed differences in NAD coverage between ESCs and NPCs are not driven by increased noise in NPCs but instead reflect biologically relevant changes in nucleolar organization (Supplementary Fig. 4a). Accordingly, the reduced NAD detection in single NPC nucleoli obtained with NoLMseq is consistent with previous Nucleolar-DamID and HiC-rDNA studies performed in NPC populations, indicating that the architecture of chromosomes surrounding the nucleolus is in a more compact and rigid form whereas in ESCs it appears more flexible and variable[20]. Moreover, these data are consistent with previous reports indicating that ESCs harbor a more open and dynamic chromatin than differentiated cells[33,34]. Consistent with this, we observed more variability in nucleolus association in ESCs than in NPCs, as indicated by Yule's Q coefficient, which provides a measure of similarity between single-cell NADs (Fig. 3e).

Next, we investigated whether genes contacting nucleoli in ESCs but detaching from nucleoli in NPCs (ESC-specific NADs, ESC$_{sp}$-NADs) could change their expression state in NPCs. We applied stringent criteria for defining ESC$_{sp}$-NADs as NADs with nucleolar CF > 25% in ESCs and <10% in NPCs. The analysis of published RNAseq data[20] revealed that genes located at ESC$_{sp}$-NADs are significantly less expressed in ESCs than in NPCs (Fig. 3f,g), suggesting that the detachment from nucleoli might facilitate gene activation. Moreover,

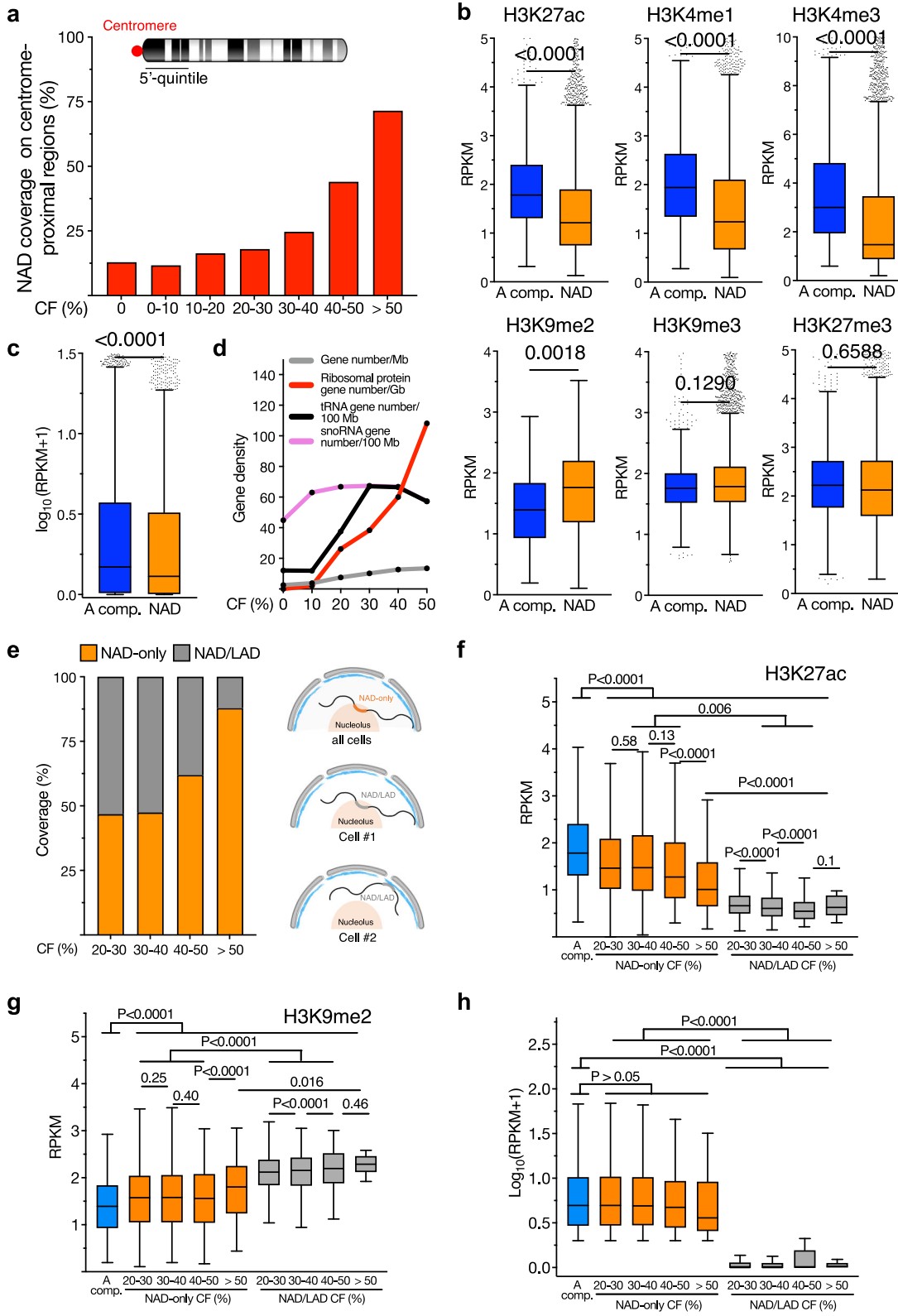

the absolute fold change in gene expression in NPCs compared to ESCs was higher for the upregulated genes, with a large fraction (60%) not expressed in ESCs (Fig. 3h, i). In contrast, genes in NPCₛₚ-NADs, common NADs, or A compartment show no significant changes in expression (Supplementary Fig. 4b). Notably, previous cell population studies using Nucleolar-DamID²⁰ failed to detect gene activation at ESCsp-NADs, highlighting the improved resolution of NoLMseq in

identifying NADs compared to cell population-based methods (Supplementary Fig. 4c). Accordingly, GO terms for these upregulated genes were significantly linked to developmental processes, in particularly nervous system development, whereas downregulated genes were mainly implicated in metabolic processes (Supplementary Fig. 4d, Supplementary Data 1). These results show that the detachment of genes from nucleoli during ESC differentiation correlates with

**Fig. 2 | NADs identified by NoLMseq correspond to repressive chromatin domains. a** NAD coverage on the centromere-proximal regions (5′-quintile) as a function of nucleolar contact frequency. **b** Levels of active histone marks (H3K27ac, H3K4me1, H3K4me3) and repressive histone marks (H3K9me2, H3K9me3, H3K27me3) at NADs relative to genomic regions located at the active A compartment (A comp.). Values are shown as average RPKM. Tukey boxplot where box limits represent the 25th and 75th percentiles. The horizontal line within the boxes represents the median. Statistical significance (*P*-values) was calculated using the unpaired two-tailed *t*-test. **c** Expression values (RPKM) of genes within the active A compartment (A Comp.) and NADs. Tukey boxplot where box limits represent the 25th and 75th percentiles. The horizontal line within the boxes represents the median. Statistical significance (*P*-values) was calculated using the unpaired two-tailed *t*-test. **d** Gene densities of ribosomal protein genes, tRNA genes, snoRNA

genes and all genes as a function of nuclear CF. **e** Left panel shows nucleolar contact frequency for NAD-only and NAD/LAD sequences. The right panel depicts how NAD-only and NAD/LAD contact nucleoli in single cells. **f–h** Levels of H3K27ac (**f**), H3K9me2 (**g**), and gene expression (**h**) at NAD-only and NAD/LAD as a function of nucleolar CF relative to the active A compartment (A comp.). at the active A compartment (A Comp.) and at NAD-only and NAD/LAD relative to nucleolar CF. Values are shown as average RPKM. Tukey boxplot where box limits represent the 25th and 75th percentiles. The horizontal line within the boxes represents the median. Statistical significance (*Q*-values) was calculated using the Benjamini and Hochberg method. All box plots depict the minimum and maximum values. The horizontal line within the boxes represents the mean value. Source data are provided as a Source Data file.

gene activation, suggesting a regulatory role of the nucleolus for the establishment or maintenance of repressive chromatin states.

## ESCs are composed of two populations with distinct nucleolar contacts

After having validated the application of NoLMseq to identify NADs in single cells at multiple levels, we started to analyze unexplored features of NADs, such as the degree of cell heterogeneity in genome organization around nucleoli. We calculated the NAD coverage for each chromosome in single cells and performed hierarchical clustering (Fig. 4a). We found two main ESC populations that differ according to the level of nucleolar contacts among chromosomes. The major differences between these two groups were in the level of rDNA-chromosome contacts with the nucleolus, which showed a mutually exclusive relationship. Large portions of rDNA-chromosomes 12 and 19 had high levels of nucleolar contacts in population 1 but not in population 2 (Fig. 4b). In contrast, rDNA-chromosomes 16 and 18 displayed higher NAD coverage in population 2 than in population 1. In the case of chromosomes not bearing rRNA genes, chromosomes 7, 11, 15, and 2 showed more nucleolar contacts in population 2 than in population 1, whereas chromosomes 3 and 5 preferentially contact nucleoli of population 1. Although ESCs are predominantly in the S phase, the potential influence of cell cycle factors cannot be entirely ruled out. Nonetheless, it is notable that the analysis of histone modifications revealed that NADs enriched in population 1 were significantly depleted of active histone marks and enriched in H3K9me2 relative to NADs of population 2, suggesting different chromatin states between these two ESC populations (Fig. 4d). To assess whether these enrichments were specific and not due to random fluctuations, we performed 1000 bootstrap randomizations of the NADs of populations 1 and 2. In none of these iterations did we observe enrichment values for H3K9me2, H3K4me1, or H3K27ac that matched or exceeded those observed in the actual data, supporting the specificity of our results (Supplementary Fig. 5a).

GO and pathway analyses showed that the specific NAD-genes of population 2 have significant enrichments in pathways linked to development and pluripotency, whereas NAD-genes of population 1 are linked to metabolism and signaling pathways (Supplementary Data 2). Thus, the presence of these two major populations could reflect the well-documented heterogeneity of ESCs cultured in serum, the conditions used in this study, characterized by a wide variety of morphologies and differential expression of pluripotency regulators and lineage specifying factors[35,36]. In particular, the expression of the pluripotency factor Nanog is heterogeneous in ESCs, giving rise to two major populations: Nanog-high and Nanog-low[37,38]. The Nanog-high population expresses markers for pluripotency and is in a more pluripotent state than the Nanog-low population. To determine whether the two ESC populations identified for their NAD profiles correlate with Nanog expression levels, we performed single-cell (sc)RNAseq. Uniform manifold approximation and projection (UMAP) displayed 7 distinct clusters of ESC populations (Fig. 4e, Supplementary Fig. 5b, c).

Nanog-high population corresponds to clusters 0, 1, 2, and 3. Next, we examined the top 30 genes, selected from a total of 282 genes of population 1 and 126 genes of population 2, that showed the highest interactions with nucleoli and that were specific for one of the two populations (i.e., NAD-genes of population 1 that have CF > 30% in population 1 and <5% CF in population 2 and vice versa for genes of population 2). Notably, a large fraction of NAD genes in population 2 (22) co-express with *Nanog* and *Sox2* in clusters 0–3, whereas only a few genes (12) from population 1 exhibit this pattern with 13 genes mainly expressed in cluster 7 (Fig. 4f). Consistently, the UMAP analysis of the top five nucleolus-interacting genes revealed that four out of five genes from population 2 share an expression profile characteristic of Nanog-high and Sox2-positive cells, while only one gene from population 1 displays a similar pattern (Fig. 4g).

These results suggest that the ESC populations 1 and 2 might possess different pluripotency states. Taken together, the data revealed structural heterogeneity in the ESC genome, resulting in two major populations distinguished by their genome organization around nucleoli, each characterized by distinct chromatin states and eventually different developmental potentials.

## Nucleolar integrity is required for genomic contacts with nucleoli

Nucleolus structure is highly dynamic and promptly responds to external stimuli that affect ribosome biogenesis and nucleolar structure. For example, downregulation of rDNA transcription by treatment with low doses of Actinomycin D (ActD) induces a spatial reorganization of the nucleolar structure, including the migration of rRNA genes to the nucleolar periphery, forming the so-called nucleolar caps, and the release of the granular component (GC) protein NPM1 from nucleoli (Fig. 5a)[39]. However, until now, it could not be investigated whether these structural changes can impact the organization of the genome around nucleoli, mainly due to the technical limitations of the current technologies to map NADs under conditions in which nucleolar integrity is affected. Since nucleoli in ESCs treated with ActD (ESCs + ActD) are still detectable with brightfield microscopy, we reasoned that NoLMseq could be applied to determine how nucleolus integrity affects chromosome contacts with nucleoli. Compared to untreated cells, the average NAD coverage in ESC + ActD significantly decreased from 15% to 10% (Fig. 5b, c, Supplementary Fig. 6a). Although NoLMseq revealed a substantial loss of nucleolar contacts across all chromosomes, the majority of NADs retained upon ActD treatment were from the rDNA-chromosomes (Fig. 5d). We validated this loss of contacts with nucleoli in ESCs+ActD by quantitative DNA-FISH, using probes targeting four NADs at chromosomes 1, 2, 5, and 19 that were previously identified and validated by Nucleolar-DamID[20] (Fig. 5e, f, Supplementary Fig. 6b). These are strong NADs since they contact nucleoli in 70–95% of ESCs (Fig. 5f). The DNA-FISH probe for the rDNA-chromosome 19 targeted a NAD sequence proximal to the rDNA locus. Accordingly, this NAD remained anchored to nucleoli upon ActD treatment, an expected result since rRNA genes remain

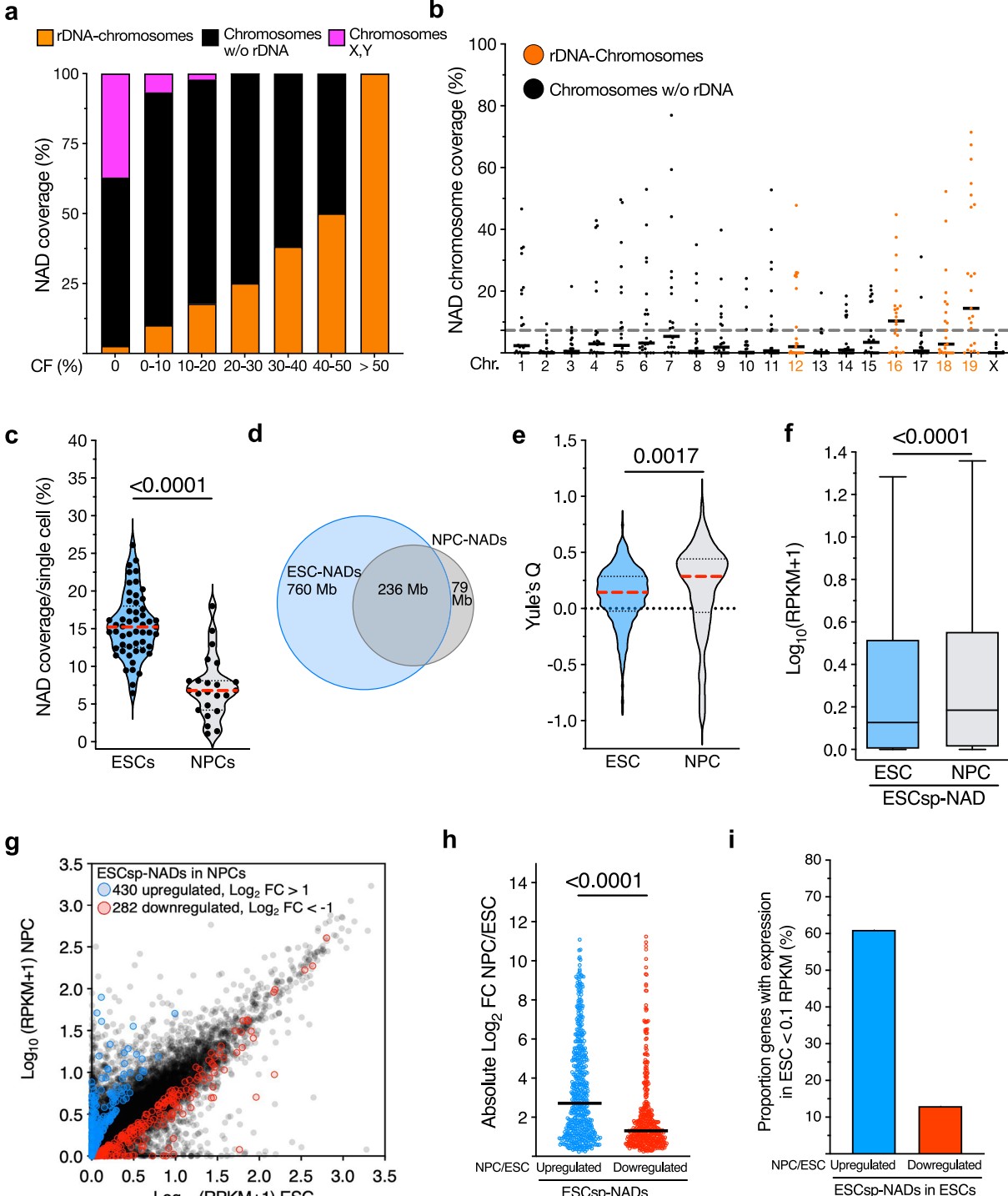

**Fig. 3 | NADs from single nucleoli of neural progenitors. a** Cumulative histogram of genome-wide nucleolar contact frequency measured from 23 microdissected nucleoli with respect to their location at chromosomes containing rRNA genes (rDNA-chromosomes), without (w/o rDNA), and X and Y chromosomes in NPCs. **b** Coverage of NADs with CF > 20% on each chromosome in single nucleoli of NPCs. The dotted gray line indicates the average NAD coverage in single cells of NPCs. Black lines indicate the mean on each chromosome. **c** Violin plots showing NAD coverage in single ESCs and NPCs. Data from ESCs were also shown in Fig. 1f. Red dashed lines indicate the median, and black dotted lines indicate the quartiles. **d** Venn diagram showing the distribution of NADs in ESCs and NPCs. **e** Violin plots showing the distribution of Yule's Q values as a measure of NAD cell−cell similarity in ESCs and NPCs. Red dashed lines indicate the median, and black dotted lines indicate the quartiles. **f** Expression levels of genes located at ESCsp-NADs in ESCs

and NPCs. Tukey boxplot where box limits represent the 25th and 75th percentiles. The horizontal line within the boxes represents the median. Statistical significance (P-values) was calculated using the paired two-tailed t-test. Box plots depict the minimum and maximum values. The horizontal line within the boxes represents the mean value. **g** Scatter plot showing the expression levels of genes at $ESC_{sp}$-NADs in ESCs and NPCs. Expression of genes significantly (P < 0.05) upregulated and downregulated is shown. **h** Plot showing the absolute $\log_2$ fold changes of upregulated and downregulated genes at $ESC_{sp}$-NADs in NPCs. Black lines indicate the median. Statistical significance (P-values) was calculated using the unpaired two-tailed t-test. **i** Proportion of low-expressing genes (<0.1 RPKM) in ESCs that are located at $ESC_{sp}$-NADs and are up- or downregulated in NPCs. Source data are provided as a Source Data file.

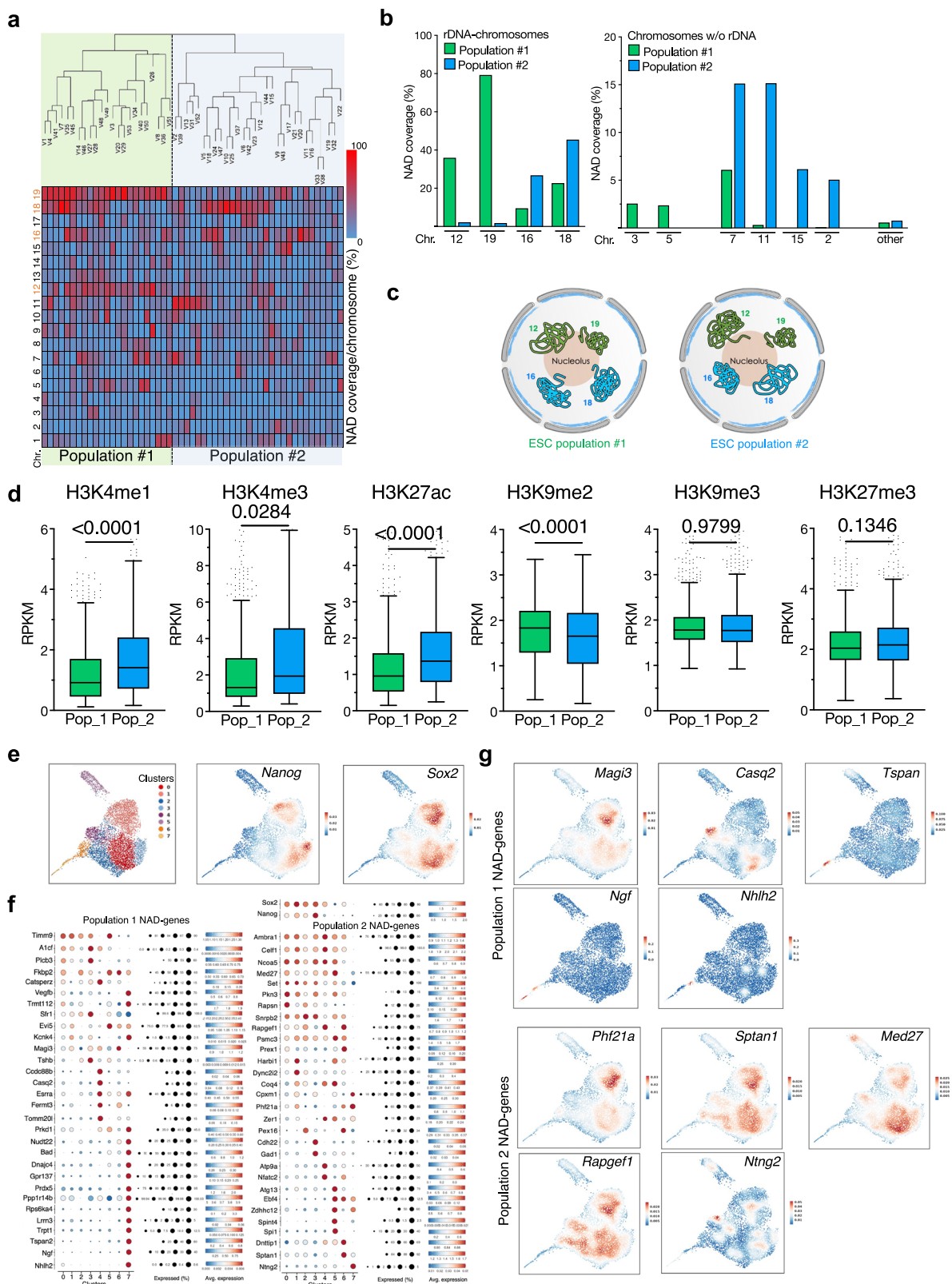

close to nucleoli by forming nucleolar caps (Fig. 5f). In contrast, DNA-FISH probes targeting NADs at chromosomes 1, 2, and 5 showed a significant detachment of NADs from the nucleoli upon ActD treatment, indicating that nucleolar integrity is crucial for NAD association with nucleoli and supporting the results obtained with NoLMseq. Because of the retention of rDNA-chromosomes to nucleoli in ESC + ActD, we reasoned that regions from rDNA-chromosomes still able to

contact nucleoli in ESC + ActD should correspond to centromere-proximal sequences, which are linearly located close to rRNA genes. Accordingly, NAD coverage of centromere-proximal sequences of rDNA-chromosomes was higher than the corresponding NAD coverage in untreated ESCs, suggesting a reorganization of rDNA-chromosomes around nucleoli (Fig. 5g). In contrast, NAD coverage of centromere-proximal sequences of chromosomes not bearing rDNA repeats

**Fig. 4 | ESCs showed two populations with distinct nucleolar contacts.**
**a** Hierarchical clustering of nucleoli according to their NAD coverage/chromo-
some. **b** NAD coverage at the indicated chromosomes in the top quartile NADs of
the two ESC populations. **c** Scheme representing the contact with nucleoli of the
rDNA-chromosomes with the nucleoli in the two ESC populations. **d** Levels of active
histone marks (H3K4me1, H3K4me3, H3K27ac) and repressive histone marks
(H3K9me2, H3K9me3, H3K27me3) at NADs enriched in ESC population (Pop) 1 or 2.
Values are shown as average RPKM. Tukey boxplot where box limits represent the
25th and 75th percentiles. The horizontal line within the boxes represents the
median. Statistical significance (P-values) was calculated using the unpaired two-
tailed t-test. Box plots depict the minimum and maximum values. The horizontal
line within the boxes represents the mean value. **e** Uniform Manifold Approxima-
tion and Projection (UMAP) of 7308 embryonic stem cells (ESCs) profiled by single-

cell RNA sequencing (scRNA-seq). Each dot represents an individual cell, colored by
Seurat-defined cluster identity (top). *Nanog* and *Sox2* expression is overlaid on the
same UMAP (bottom). Heatbar indicates normalized gene expression levels. **f** Dot
plot showing the expression patterns of the top 30 genes with the highest nucleolar
interactions specific to populations 1 and 2, as well as pluripotency markers *Nanog*
and *Sox2* across Seurat-defined ESC clusters. Dot size indicates the percentage of
cells expressing the gene within each cluster, and color intensity reflects the
average expression level. **g** UMAP projection showing expression of the top five
genes with the highest interactions with nucleoli and that were specific for popu-
lation 1 or 2. Gene expression is overlaid on the UMAP as feature plots. Heatbar
indicates normalized expression levels. Source data are provided as a Source
Data file.

---

showed a significant reduction compared to NADs of untreated cells,
further indicating that the anchoring of these sequences to nucleoli
depends on nucleoli's integrity. All these results indicate that nucleolar
integrity is an important determinant for chromosome organization
around nucleoli. To assess whether the detachment of NADs from
nucleoli upon nucleolar stress affects gene expression, we performed
quantitative RNAseq spiked-in with *ERCC* control RNAs (Fig. 5h).
Treatment with ActD for 24 h significantly altered the expression of
5567 genes (log$_2$ fold change ≥1; $P$ < 0.05), of which 3235 genes were
upregulated and 2332 were downregulated compared to control cells.
To determine whether ActD treatment affects the expression of genes
at NADs, we analyzed genes in high-confidence NAD-only regions in
ESCs (CF > 30%) (Fig. 5i). We found that 17% of these NAD-genes (507)
were upregulated upon ActD treatment, whereas only 9% (260) were
downregulated. In contrast, genes at non-NADs (CF < 20%) showed a
similar number of up- and downregulated genes. These results suggest
that the loss of nucleolar integrity might play a significant role in the
derepression of NAD-genes. The upregulated NAD-genes were linked
to signaling pathways involving calcium, PI3K-Akt, and p53, the latter
being well known to be activated by RNA Pol I inhibition upon ActD
treatment[40,41] (Supplementary Fig. 6c, Supplementary Data 3). We also
found that about half of the upregulated NAD-genes remain in contact
with nucleoli (Fig. 5i) and, accordingly, the majority of them (71%) were
located at rDNA-chromosomes. These results suggest that the
nucleolus may also lose its repressive properties under these nucleolar
stress conditions, although further investigation is required to fully
understand these properties.

Together with the reactivation of NAD genes during ESC differ-
entiation into NPCs upon loss of nucleolar contact, these results are
consistent with the possibility that the nucleolus is linked to gene
repression, although its direct regulatory role remains to be fully
established (Fig. 5j). Moreover, the data demonstrate that NoLMseq is
a unique and only genomic method for studying the role of nucleolus
in genome organization under nucleolar stress conditions that affect
nucleolar integrity.

**Genomic contacts with nucleoli are predominantly mono-allelic**
Recent data have started to detect allelic asymmetry in gene expres-
sion and 3D-genome organization[42,43]. So far, however, all studies on
NADs were performed using cells whose maternal and paternal chro-
mosomes could not be distinguished from each other, as in the case of
the E14 ESC line that derives from a fully inbred homozygous mouse
strain. Consequently, the information on NAD coverage has been
based on the assumption that both alleles contact nucleoli. To deter-
mine the allelic distribution of NADs in single nucleoli, we applied
NoLMseq to the male mouse F123-ESC line that was derived from the F1
generation of two fully inbred homozygous mouse strains: *Mus mus-
culus castaneus* (CAST, paternal chromosomes) and *Mus musculus
domesticus* S129S4/SvJae (S129, maternal chromosomes)[44]. The F123
DNA sequence has been well characterized and contains a high density
of single-nucleotide variants (SNVs), with an average of 1 SNV every 124

nucleotides across autosomes[45]. We analyzed NAD profiles from 50
microdissected nucleoli of F123-ESCs by aligning them against the F123
genome to identify allele-specific NADs (i.e., NAD$_{S129}$ and NAD$_{CAST}$)
(Fig. 6a). The vast majority (93%) of total NADs identified in F123-ESCs
corresponded to NADs identified in E14-ESCs, indicating that genomic
contacts with nucleoli are generally conserved between cell lines with
the same developmental stage but from different strains (Supple-
mentary Fig. 7a). NADs of F123-ESCs showed a slightly lower genome
coverage (11.3%, calculated to haploid genome) relative to 15% NAD
coverage measured in E14-ESCs (Fig. 6B). Surprisingly, we found that a
large fraction of NADs (86%) are mono-allelic (Fig. 6a–c). Monoallelic
NADs were equally distributed among S129 and CAST chromosomes
(Supplementary Fig. 7b). Notably, rDNA-chromosomes have about 10-
fold higher proportion of biallelic NADs compared to non-rDNA
chromosomes (Supplementary Fig. 6c). The distinction between
monoallelic and biallelic NADs was also evident from nucleolus CF
analysis, which showed that biallelic NADs have a higher CF than
monoallelic NADs whereas NAD-only and NAD/LAD regions display
similar content (Supplementary Fig. 7d). We validated these results by
DNA–FISH for two strong NADs, one located at chromosome 5 and
another located at the rDNA-chromosome 19 and close to the rDNA
locus (Fig. 6d). The NAD of chromosome 5 showed monoallelic con-
tacts in almost all the cells. Remarkably, even the NAD of chromosome
19, located close to the rDNA locus, showed mono-allelic contacts with
nucleoli in 50% of the cells, supporting the results of NoLMseq. This
prevalent mono-allelic NAD distribution resulted in a much lower NAD
coverage per cell (5.6%, 297 Mbp) compared to the previously assumed
bi-allelic distribution (Fig. 6e). To determine whether mono- and bi-
allelic NADs reflect differences in chromatin modifications, we per-
formed ChIPseq analysis for H3K9me2, H3K27ac, and H3K4me3 in
F123-ESCs (Fig. 6f). We found that bi-allelic NADs are in a more
repressed chromatin states than mono-allelic NADs since they display
reduced levels of H3K27ac and H3K4me3 and higher levels of
H3K9me2, the latter the repressive histone mark characterizing NADs
in ESCs[20]. Similarly, RNAseq analyses showed a significantly lower
expression of genes located at bi-allelic NADs (Fig. 6g). Among genes
located at bi-allelic NADs, we found cell type specific transcription
factors, such as *Gata6* and *TAF4b*, as well as genes linked to metabolic
pathways and neuroactive ligand-receptor interactions (Supplemen-
tary Fig. 7e). These results further indicate that high-frequency contact
with nucleoli, even at the allelic levels, results in a more repres-
sive state.

Next, we asked whether the mono-allelic distribution of NADs
might be parent-specific. We applied stringent criteria to assign
parent-specific NADs, considering, for example, that a specific
NAD$_{S129}$ should have >25% nucleolar CF with the S129 genome and
<10% nucleolar CF with the CAST genome (Fig. 6h). However, very
few NADs (1% on both alleles) showed a parent-specific distribution.
This indicates that, generally, in one cell, it is the paternal allele
contacting nucleoli while in another cell it is the maternal
allele (Fig. 6h).

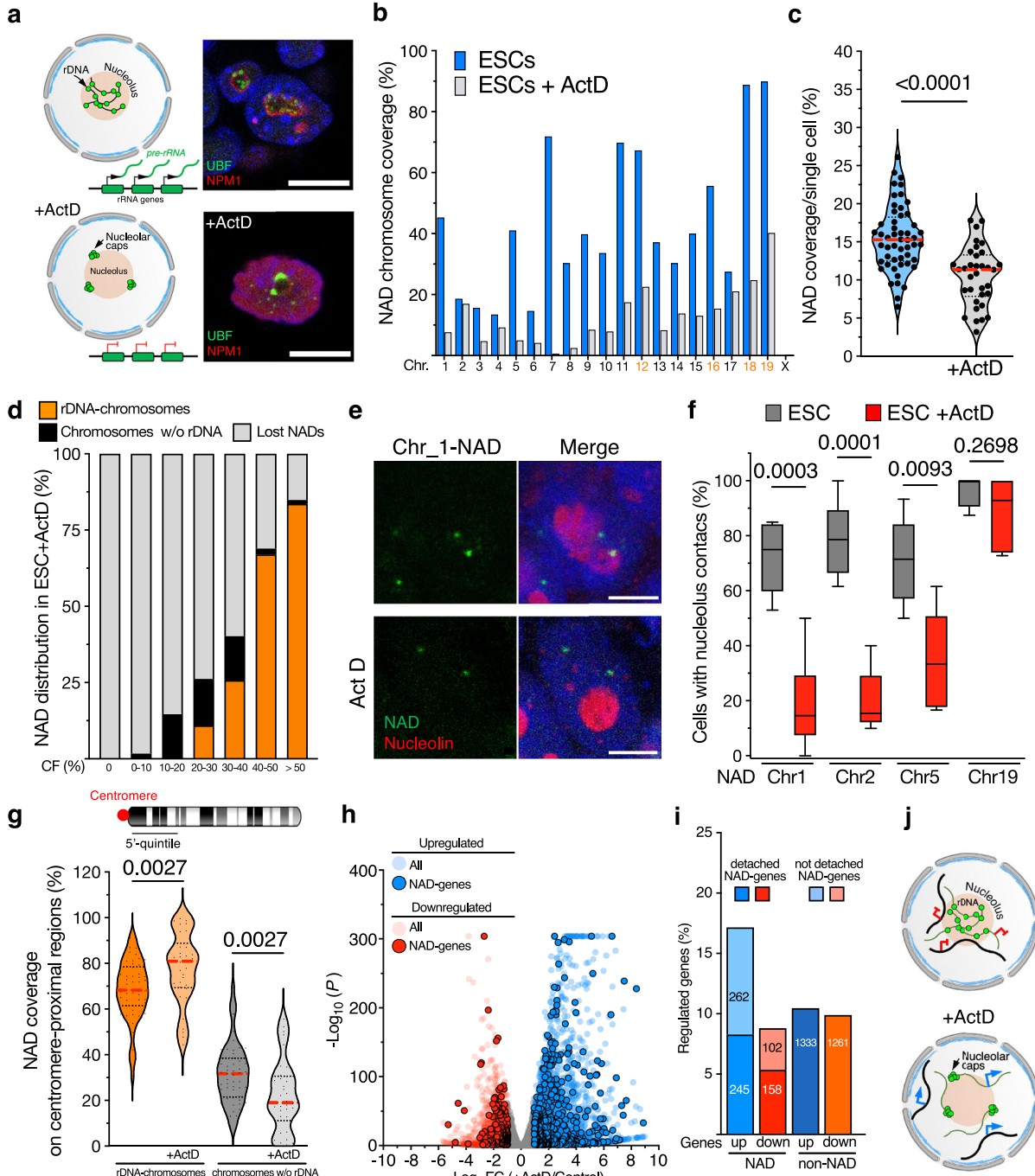

**Fig. 5 | Nucleolar integrity is required for genomic contacts with nucleoli. a** Left panel, alterations in nucleolus structure upon Actinomycin D (ActD) treatment, showing the re-organization of rRNA genes, which form nucleolar caps. Right panel, immunofluorescence images of nucleolar markers UBF and NPM1. Scale bar is 9 μm. **b** Histogram showing NAD coverage/chromosome average from single nucleoli of untreated ESCs (blue) and 34 microdissected nucleoli from ActD-treated ESCs (gray). **c** Violin plots showing NAD coverage in single untreated ESCs (blue) and ActD-treated ESCs (gray). Data from ESCs were also shown in Fig. 1f. Red dashed lines indicate the median, and black dotted lines the quartiles. Statistical significance (*P*-value) was calculated using the unpaired two-tailed *t*-test. **d** Distribution of NADs in ActD-treated ESCs as a function of nucleolar CF. **e** Representative immuno-FISH images for NAD at chromosome 2. Nucleolin serves as a nucleolar marker. Scale bar is 5 μm. **f** Box plots represent quantifications of immuno-DNA-FISH analyses showing the percentage of cells displaying NADs contacting nucleoli. Error bars represent s.d. Statistical significance (*P*-values) from

three independent experiments was calculated using a two-sided Mann–Whitney test. Box plots depict the minimum and maximum values. The horizontal line within the boxes represents the mean value. **g** NAD coverage at the centromere-proximal regions (the first 5'-quintile) between rDNA–chromosomes and chromosomes without (w/o) rDNA. rDNA on untreated ESCs and ESCs treated with ActD. Red dashed lines indicate the median and black dotted lines the quartiles. Statistical significance (*P*-value) was calculated using the unpaired two-tailed *t*-test. **h** Volcano plot showing fold change (log₂ values) in transcript levels of ESC + ActD and control ESCs. Gene expression values of three replicates were averaged and selected for log2 fold changes ≥ 1 and adjusted (adj.) *P* < 0.05. **i** Proportion of NAD-only genes (CF < 30%) and non-NAD genes (CF < 20%) in ESCs that become up- and downregulated in ESCs. **j** Schematic representation of alterations in genomic contacts with nucleoli and gene expression following ActD treatment. The upregulation of NAD genes and the formation of nucleolar caps are shown. Source data are provided as a Source Data file.

## Discussion

We described NoLMseq, a method able to measure genome organization around nucleoli in single cells. LCM has been widely used in various omics approaches for isolating single cells or specific tissue regions[23–27]. NoLMseq has adapted this technology to study subcellular genomic compartments, specifically the nucleolus, enabling precise examination of the genome architecture surrounding the nucleolus in single cells.

Relative to previous NAD methods, NoLMseq not only detects NADs in single nucleoli, but it can also be applied to study genome organization around nucleoli under conditions of nucleolar stress, which profoundly alter nucleolar structure[6–9] and that could not be examined with current methods, making it elusive how genome structure around nucleoli responds to nucleolar stress. The data revealed a profound reorganization of the genome around "stressed" nucleoli, such as an increase in nucleolar contacts of centromere-proximal sequences of rDNA–chromosomes, while the rest of the chromosomes lose their contact. The genomic data obtained by NoLMseq are also consistent with recent image analyses showing that perturbation of nucleolus structure upon downregulation of the RNA polymerase I subunit RPA194 reduced centromere-nucleolus interactions[46]. These results indicate that nucleolar structure and integrity play an important role in genome organization, and future studies will address alterations in chromosomal interactions with nucleoli under pathological stress conditions, impacting ribosome biogenesis, and how these perturbations are caused and their functional significance. The results also showed that NADs are very dynamic and NoLMseq represents a unique technique able to capture NADs at the present state rather than averages of the past and present states, as in the case of DamID, as recently discussed[47].

The detection of NADs in single nucleoli using NoLMseq has proven to be highly precise compared to previous methods and capable of providing unexplored insights into genome organization. For example, NoLMseq determined that active genes, such as ribosomal protein genes and tRNA genes, also have high-frequency contacts with nucleoli. Interestingly, previous imaging analyses in living or fixed yeast cells reported the proximity of some ribosomal protein genes and tRNA genes to nucleoli[48,49], suggesting that the nucleolus could act as a compartment for the regulation of the expression of these key components of the translation machinery. The presence of some active genes at nucleoli is also consistent with a recent study showing that the localization of genes at the repressive B compartment or proximal to nucleoli does not necessarily preclude transcription[50].

The accuracy of NoLMseq in detecting NADs relative to cell population studies is also shown by the identification of $ESC_{sp}$-NADs that contain genes activated in NPCs, which could not be detected in previous cell population studies using Nucleolar–DamID[20]. One explanation for these results is that while nucleolar DamID allows for the classification of genomic regions as associated or not associated with the nucleolus, it lacks the sensitivity to quantify the degree of this association. In contrast, NoLMseq enables the quantification of nucleolar CF and the use of stringent criteria to robustly define NADs, as in the case of $ESC_{sp}$-NADs, defined as high-confidence NADs in ESCs that were consistently excluded from the nucleolus (>25% CF in ESCs, <10% CF in NPCs). Thus, this quantitative resolution provides a more nuanced view of nucleolar genome organization and enables the detection of gene expression changes that were only partially captured with Nucleolar–DamID. These results are consistent with the possibility that the nucleolus might be linked to gene repression.

Because of the single-nucleolus detection, NoLMseq revealed that ESCs display a certain heterogeneity in genomic contacts, with nucleoli showing two major cell populations with distinct chromatin features and potentially associated with different developmental states. These results are also consistent with recent data showing heterogeneity of nuclear organization between individual single ESCs even when cultured with a '2 inhibitor' (2i) cocktail that is known to induce homogenous expression profiles across single cells[51,52]. Finally, the application of NoLMseq in the hybrid F123-ESC line showed that genomic contacts with nucleoli are mainly monoallelic. This result clearly provided a more accurate value of NAD coverage, which was previously calculated based on either a haploid genome or by assuming that all NADs are bi-allelic in the diploid genome. The average NAD coverage of 5.6% (i.e., ca. 297 Mbp) per cell obtained in the F123-ESC line is much lower than the 30% NAD coverage measured in ESC population studies using both purified nucleoli and Nucleolar–DamID methods[18,20]. Although this calculated NAD coverage value does not include repetitive sequences such as major and minor satellites, which are known to contact nucleoli but are not present in the reference genome, it clearly represents a more realistic scenario than having one-third of chromosomes contacting nucleoli. Moreover, the data showed that bi-allelic NADs have high nucleolar CF and are in a more repressive state than monoallelic NADs, further supporting the efficacy of NoLMseq in identifying NADs and the nucleolus acting as a repressive compartment.

A limitation of NoLMseq is that it can be applied only to cells with one nucleolus, like mESCs, since the current state of laser technology does not allow the microdissection of very small cellular organelles. However, there are many biomedically relevant cell types that have one large nucleolus, such as hematopoietic stem cells, adult stem cells, and T cells, to name a few. On the other side, NoLMseq offers several unique advantages over other methods, as it is the only method likely to be applicable under conditions of nucleolar stress, as demonstrated by the analysis of ESCs treated with ActD.

Together, the results demonstrate that NoLMseq not only accurately measures chromosome contacts around single nucleoli but also provides novel insights into genome organization around nucleoli that could not be detected by previous studies. We predict that NoLMseq will not only be a critical tool to study genome organization around the nucleolus in biological populations but also to study how the 3D-genome organization responds to nucleolar stress in disease states and understand the functional consequences of these alterations.

## Methods

### Cell lines

One hundred and twenty-nine mouse embryonic stem cells (E14 line) were cultured in serum medium containing Dulbecco's modified Eagle's medium + Glutamax (Life Technologies), 15% FCS (Life Technologies; Cat no. 10270106 FBS South American), 1× MEM NEAA (Life Technologies), 100 μM β-mercaptoethanol, recombinant leukemia inhibitory factor, LIF (Polygene, 1000 U/ml), 1× penicillin/streptomycin (Life Technologies). ESCs were seeded at a density of 50,000 cells/cm$^2$ in culture dishes (Corning CellBIND surface) coated with 0.1% gelatin without a feeder layer. Propagation of cells was carried out every 2 days using enzymatic cell dissociation. ESCs were treated with Actinomycin D (10 ng/ml) for 24 h before downstream analyses.

Neural progenitor cells were generated from ESCs, according to a previously established protocol[53]. In brief, differentiation used a suspension-based embryoid bodies formation (Bacteriological Petri Dishes, Bio-one with vents, Greiner). The neural differentiation media (DMEM, 10% fetal calf serum, 1× MEM NEAA, 2 mM Pen/Strep, β-mercaptoethanol, and sodium pyruvate) were filtered through 0.22 μm filters and stored at 4 °C. During the 8-day differentiation procedure, the media were exchanged every 2 days. In the last 4 days of differentiation, the media were supplemented with 2 μM retinoic acid to generate neural precursors that are Pax-6-positive radial glial cells.

F123-ESCs (mouse male, hybrid cell line, F1 S129/Jae and Cast mouse cross)[44] were cultured on a layer of mitotically inactivated feeder murine embryonic fibroblasts under standard conditions (DMEM, supplemented with 15% KSR, 1× Glutamax, 10 mM non-essential amino acids, 50 μM beta-mercaptoethanol, 1000 U/ml LIF).

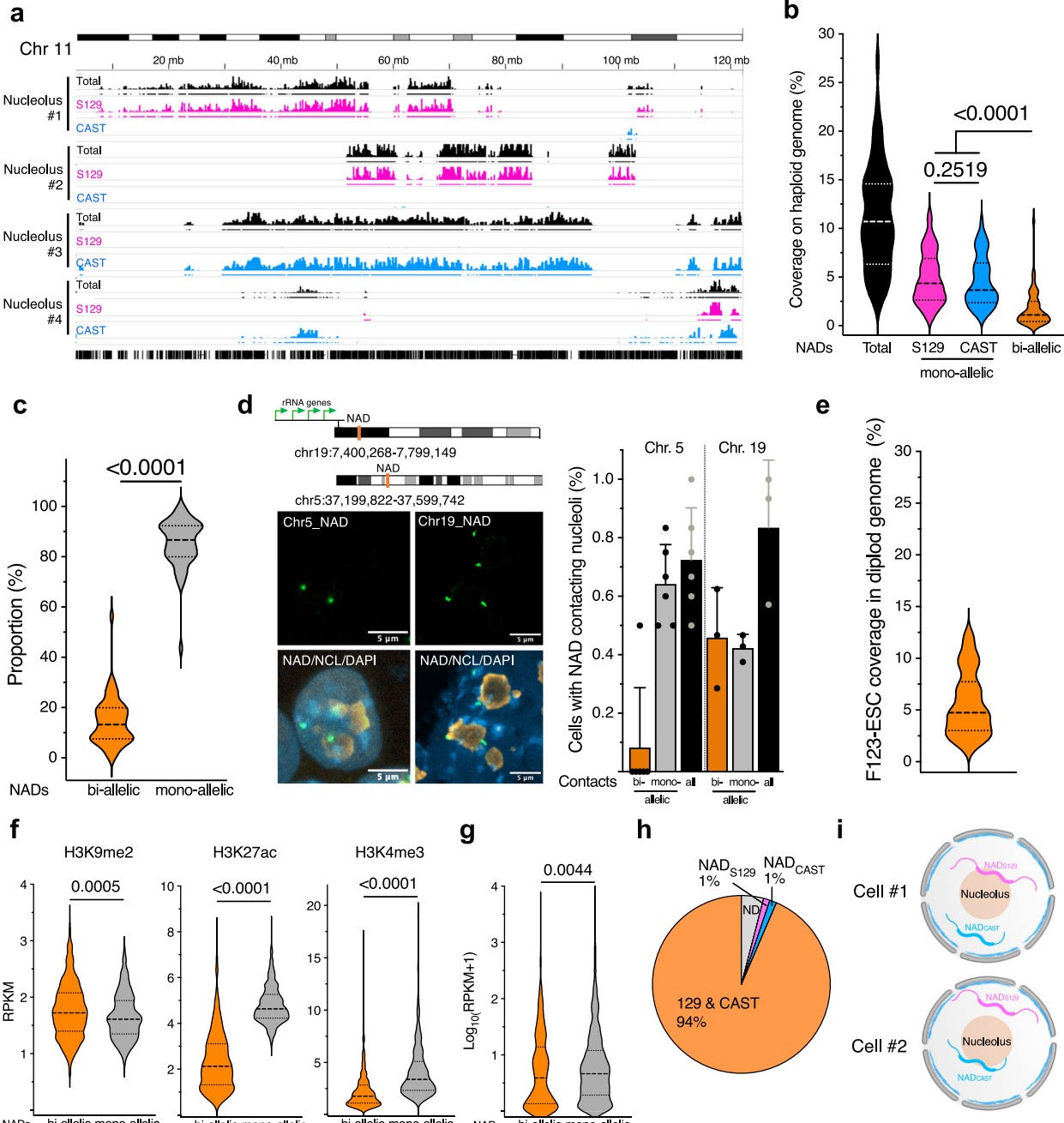

**Fig. 6 | Genomic contacts with nucleoli are mono-allelic. a** NoLMseq tracks displaying NAD sequences from four microdissected single nucleoli from hybrid F123-ESCs (S129/CAST) on the total mm10 genome, maternal S129 genome and paternal CAST genome. Bars below the tracks indicate NADs called on respective genomes. **b** NAD coverage from single nucleoli calculated over the haploid genome. Median and quartiles are shown. Statistical significance (*P*-value) was calculated using the unpaired two-tailed *t*-test. **c** Proportion of bi-allelic and mono-allelic NADs in single nucleoli. Black dashed lines indicate the median, and black dotted lines indicate the quartiles. Statistical significance (*P*-value) was calculated using the unpaired two-tailed *t*-test. **d** Left panel shows DNA-FISH images of an NAD located on chromosome 19. The right panel is the quantification of bi-allelic and mono-allelic contacts with nucleoli. Data are from three independent experiments. Scale bar is 5 μm. **e** NAD coverage from single nucleoli over the diploid genome. The black dashed line indicates the median, and the black dotted lines indicate the quartiles. **f, g** Levels of H3K9me2, H3K27ac, and H3K4me3 (**f**) and gene expression (**g**) at bi-allelic and mono-allelic NADs. Values are shown as average RPKM. Violin plots where black dashed lines indicate the median and black dotted lines indicate the quartiles. Statistical significance (*P*-values) was calculated using the unpaired two-tailed *t*-test. **h** Pie chart showing the amounts of paternal (NAD_{CAST}) and maternal (NAD_{S129}) NADs. **i** Model showing that mono-allelic distribution of NADs at nucleoli of single cells is generally not parent-specific. Source data are provided as a Source Data file.

Media was changed every 24 h. Before harvesting, mESCs were passaged onto feeder-free ESGRO-gelatin (EMD Millipore SF008). coated plates for at least 2 passages to remove feeder cells. Cells were harvested after approximately 48 h at 70–80% confluency. F123-ESCs were a gift from Bing Ren, University of California, San Diego.

**NoLMseq**

For laser-microdissection, cells were grown on ESGRO-gelatin-coated and UV-irradiated PET membrane slides (1.4 μm thick, RNase-free, MMI, 50102) for 24 h. Cells were washed twice with 1× PBS and fixed with EM-grade freshly-depolymerized 3% PFA in 1× PBS for 30 min. The

slides were then placed in 70% ethanol for 1 min and air dried. The slides were then washed with filtered ultra-pure $H_2O$ ($3 \times 1$ min each) and air dried. Single nucleoli were laser microdissected with the MMI Laser Capture Microdissection Nikon microscope (Molecular Machines & Industries) into single 0.2 ml tubes (transparent MMI caps, 50208). Nucleoli were identified by bright-field imaging. A 0.2 ml cap filled with transparent adhesive material was lowered onto the cells. A laser was used to cut the PET membrane surrounding each nucleolus and the cut PET membrane section along with the single nucleolus was lifted off by the sticky cap. Each single nucleolus was collected in a separate cap and stored for further processing. Nucleoli identified using bright-field microscopy were initially focused using Köhler illumination to achieve optimal contrast for accurate visualization of the nucleoli. To ensure precision, the stage was adjusted with camera-stage alignment, maintaining an inclination of less than ±0.1°. Paraxial lens offset was employed to align the optical axes of all objectives. Laser alignment was calibrated by firing a single shot, adjusting the pixel center of the imaging system to coincide with the laser beam center. Laser cutting parameters, such as cut velocity, laser focus, and laser power, were then optimized. The cut velocity was set at 20 μm/s, laser power at 0.001% of maximum capacity, and the focus was calibrated for each cut to achieve a cut diameter of 0.25 μm thickness, ensuring reproducibility. Prior to nucleolus excision, the laser was refocused on an empty area of the slide, where a cut similar in size to the nucleolus was drawn. The laser's inner ring was positioned 0.25 μm from the edge of the drawn shape. When the cap was lowered onto the membrane with the sample, any membrane displacement (typically a few microns) was corrected using the Cap Z offset. To maintain consistent focus during slide movement, the slide surface focusing function of the microscope was employed to account for potential topological differences in the membrane. Upon identifying a cell with a distinct nucleolus, a 0.2 mL cap containing a transparent adhesive was lowered onto the sample. The inner ring of the laser used to cut the PET membrane is positioned approximately 0.25 μm from the outer edge of the nucleolus. The excised PET membrane, along with the nucleolus, was lifted off using the adhesive cap. The cap was lowered onto an empty area of the membrane to check that the sample was correctly excised. Any caps with improper cuts or those lacking a clearly identifiable nucleolus were discarded. Only caps containing correctly isolated nucleoli were retained for further processing and analysis. After collection, a new cap was added, and the entire focusing and cutting process was repeated to ensure consistency across samples. Each nucleolus was collected in a separate cap for further processing.

Whole-genome amplification (WGA) was carried out using the MALBAC single-cell WGA kit (YK001A) following the manufacturer's recommendations with minor modifications. The entire WGA reaction was carried out in the same 0.2 ml tube with a single nucleolus. Another tube, with an empty section, was used as a negative control, whereas 30 pg of whole cell DNA was used as a positive control for the WGA reaction. The cell lysis reaction was first carried out by using a heated lid at 50 °C for 50 min, and then samples were collected by briefly spinning and proceeding with the cell lysis reaction again at 50 °C for 50 min, followed by inactivation at 80 °C for 10 min. The reaction is then cooled to 4 °C. The lysis step is followed by a pre-amplification step, which involves the quasilinear amplification using random primers at 20 °C progressively with 10 °C steps till 70 °C. After cooling, the samples were subjected to a final amplification step by 20 cycles of PCR.

WGA-amplified DNA was purified using a Qiagen MinElute PCR Purification Kit and eluted in 140 μl ultrapure library-grade $H_2O$. The amplified nucleolar DNA was checked on a 2% agarose gel to select positive samples showing smears corresponding to amplified DNA similar to the positive control. The positive samples were further checked for the presence of rDNA and the absence of the Tubulin gene by PCR on an agarose gel. The samples that passed this quality control check were then used for library preparation and sequencing. In total, 120 μl of the reaction was first sonicated in split-cap tubes by Covaris E220 to obtain fragments around 300–500nt in length (Peak power−75, Duty factor−20, cycle/bursts−1000, duration−35 s). The samples were then purified and concentrated with AMPure XP beads (Beckman Coulter, A63881) and used for Illumina library using the NEBNext ultra II kit (NEB E7600). Briefly, NoLMseq samples (10–100 ng) were end-repaired and polyadenylated before the ligation of Illumina-compatible adapters. The adapters contain the index for multiplexing. Library concentrations were estimated using a Qubit® 4 fluorometer (Thermo Fisher Scientific) and the 4200 TapeStation system, and libraries were pooled together in batches averaging 40 samples. Each library pool was sequenced in single-end 100 bp in one lane of Novaseq6000 to get an average of 10 million reads per library.

For data analysis, NoLM-seq reads were aligned to the mouse mm10 reference genome using Bowtie2 (version 2.5.0[54]) with default parameters. PCR duplicates were removed using samtools[55]. Positive NAD bins were called using SICER2[56], using the sequenced gDNA file as the control file and the parameters $w = 100{,}000$, $g = 100{,}000$, fdr = 0.0001. Bigwig files were created from the bam files using bamCoverage from deeptools[57]. For F123-ESCs, an N-masked genome was created using the mm10 reference genome and the Variant Call *Format* (VCF) file from the mouse genome project containing the SNP information for all strains. The SNPsplit package[58] was used to read the VCF file and extract the SNP information for the S129/SvJae and CAST strains and N-mask for the mm10 genome. The reads were aligned using Bowtie2 to this N-masked reference genome using default parameters. SNPsplit was then used to tag the SNPs for S129/SvJae and CAST alleles in the bam file and sort the haplotype-specific bam files for both alleles. bamCoverage and Sicer2 positive NAD calling were performed independently on the allele-specific bam files. Integrative Genome Viewer (IGV, version 2.15.4)[59] was used to visualize and extract representative single-nucleolus NoLMseq bigwig and bed tracks.

Nucleolar contact frequency was calculated at a 100 kb resolution. We defined NADs as sequences with a >20% nucleolar contact frequency. Our sequencing results show that ~15% of the genome is recovered per single nucleus. Given the NAD volume, this coverage value indicates that one-third of the recovered sequences upon microdissection of one single cell may arise from non-NADs. Thus, in each single microdissected sample, approximately one-third of the captured sequences would originate from non-NAD regions. By integrating the data from 53 microdissected cells using a CF threshold of 20%, the probability that a non-NAD region is falsely classified as a NAD is only 0.000144%. Since the mouse genome comprises 27,904 100 kb bins, the expected number of false-positive calls in the data is 0.04, less than a single bin. CF threshold was calculated using the cumulative distribution function (CDF) $P(X \le k) = \sum i = 0 k \binom{n}{i} p^i (1-p)^{n-1}$ where "n" is the number of single nucleoli samples, "*p*" is the probability that a false-positive region is present in the sample, "*k*" is the contact frequency threshold. For the hierarchical clustering of NADs, NAD chromosome coverages in single nucleoli were used for clustering single nucleoli based on divisive clustering (Diana) using the "cluster" package (version 2.1.6) in R.

For randomization, we generated 53 randomized control datasets, each matched in genome-wide coverage to one of the 53 ESC NoLM-seq samples. For each dataset, we randomly assigned 100 kb genomic bins as either "NAD" or "non-NAD," matching the total number and proportion of NAD-classified bins as those observed in the corresponding experimental samples. The contact frequency distribution was calculated similarly to ESC NoLM-seq samples. Proportions of regions from rDNA-chromosomes, non-rDNA chromsomes and XY-chromosomes over contact frequencies were plotted using Graphpad Prism (version 10.5.0). Randomized control datasets were generated for NPCs in a similar manner.

Bootstrapping analysis was performed in R using 1000 iterations to randomly sample NADs from the NADs of populations 1 and 2. Enrichment for each histone mark (H3K27ac, H3K4me1, H3K4me3, H3K9me2, H3K9me3, and H3K27me3) was calculated and tested for significance. The empirical *p*-value was computed as the proportion of bootstrap iterations in which the enrichment difference was equal to or greater than the observed difference between the two subpopulations.

Autocorrelation analysis was performed using a custom Python script on the NAD profiles from single nucleoli of embryonic stem cells (ESCs) and neural progenitor cells (NPCs). The autocorrelation function (ACF) was computed for each sample using Pearson correlation coefficients at increasing lag distances (1–50 bins). To quantify the rate of autocorrelation decay, we fitted an exponential function of the form $ACF(x) = a \cdot e^{-bx}$, where *a* represents the initial autocorrelation and *b* is the decay constant, which reflects how rapidly NAD associations are lost over increasing genomic distances. Data visualization was carried out using Matplotlib (https://github.com/matplotlib/matplotlib). ACFs and their corresponding exponential fits were plotted for both ESCs and NPCs, with fitted parameters (amplitude *a* and decay constant *b*).

### RNAseq

Total RNA was purified with TRIzol reagent (Life Technologies). The quality of the isolated RNA was determined with a Qubit® 4 fluorometer (Life Technologies, California, USA) and 4200 TapeStation system (Agilent, Santa Clara, California, USA). Only those samples with a 260 nm/280 nm ratio between 1.8 and 2.1 and a 28S/18S ratio within 1.5–2 were further processed. The TruSeq Stranded mRNA (Illumina, Inc., California, USA) was used in the succeeding steps. Briefly, total RNA samples (100–1000 ng) were polyA-enriched and then reverse-transcribed into double-stranded cDNA. The cDNA samples were fragmented, end-repaired and adenylated before ligation of TruSeq adapters containing unique dual indices (UDI) for multiplexing. Fragments containing TruSeq adapters on both ends were selectively enriched with PCR. The quality and quantity of the enriched libraries were validated using the Qubit® 4 fluorometer. The product is a smear with an average fragment size of approximately 260 bp. Libraries were normalized to 10 nM in Tris-Cl 10 mM, pH 8.5, with 0.1% Tween 20. The NovaSeq X (Illumina, Inc., California, USA) was used for cluster generation and sequencing according to standard protocol. Sequencing was paired-end at 2 × 150 bp.

The quality of the reads was checked by FastQC, a quality control tool for high-throughput sequence data[60]. The quality of the reads was increased by applying: a) SortMeRNA[61] (version 2.1) tool to filter ribosomal RNA; b) Trimmomatic[62] (version 0.40) software package to trim the sorted (a) reads. N-masked genome was created using the mm10 reference genome and the VCF file from the mouse genome project containing the SNP information for all strains. The SNPsplit package[58] was used to read in the VCF file and extract the SNP information for the S129/SvJae and CAST strains and N-mask the mm10 genome. The sorted (a), trimmed (b) reads were mapped against the Nmasked mouse genome (mm10) using the default parameters of the STAR (Spliced Transcripts Alignment to a Reference, version 2.6[63]). SNPsplit was then used to tag the SNPs for S129/SvJae and CAST alleles in the bam file and sort the haplotype-specific bam files for both alleles. For each gene, reads per kilobase per million (RPKM) were calculated as $RPKM = (10^9 * C)/(N * L)$, where *C* is the number of reads mapped to a gene, *N* is the total mapped reads in the experiment, and *L* is the gene length in base-pairs for a gene[64].

Cells were treated with Actinomycin D (10 ng/mL) for 24 h, after which $1 \times 10^6$ cells were collected for RNA purification using the Zymo RNA extraction kit (Zymo Research). Three biological replicates were prepared for both ActD-treated and untreated conditions. RNA isolation was performed according to the manufacturer's instructions. All samples were then spiked with ERCC control RNAs (Thermo Fisher Scientific), following the manufacturer's recommendations.

Library preparation was carried out using the TruSeq Stranded mRNA kit (Illumina, Inc., California, USA) as described previously. The absolute abundance of mRNA transcripts was assessed using the ERCC92 RNA spike-in control[65]. Most of the 92 ERCC transcripts showed consistent counts between untreated and ActD-treated samples. Reads were aligned to the mouse genome using STAR and RPKM values as described above.

### Single-cell RNAseq

ESCs were cultured as described above, and a single-cell suspension was prepared by dissociating the cells by treatment with Accutase at 37 °C for 3 min. The reaction was quenched with serum-containing medium, and cells were washed twice with PBS supplemented with 0.04% bovine serum albumin (BSA). The concentration of the single cell suspension was assessed using an automated cell counter (LUNA-FX7, Logos) with a fluorescent dye (AO/PI). Library preparation was conducted using the 10× Chromium Next GEM Single Cell 3' Kit v3.1. Approximately 10,000 cells were loaded onto the Chromium Next GEM Chip G. Library preparation followed the manufacturer's guidelines outlined in the User Guide (CG000315, Rev F). For sequencing, the resulting libraries were processed on one lane of an Illumina NovaSeq X Plus 10 billion flow cell with a 150 bp paired-end read configuration. Read alignment was performed using STAR against the mouse GRCm39 reference genome. Count matrices were generated with CellRanger (10× Genomics, v8.0.0) using the GRCm39 reference genome and GENCODE M31 gene annotations. Analysis yielded an average of approximately 55,544 reads and 30,658 unique molecular identifiers (UMIs) per cell, across a total of 7304 recovered cells. All downstream single-cell analyses were conducted using the Seurat framework (v5.1). Data visualization was performed using the scRNA shiny app (https://github.com/ucdavis-bioinformatics/scRNA_shiny).

### ChIPseq

ChIP analysis was performed as previously described[66]. Briefly, 1% formaldehyde was added to cultured cells to cross-link proteins to DNA. Isolated nuclei were then lysed with lysis buffer (50 mM Tris-HCl, pH 8.1, 10 mM EDTA, pH 8, 1% SDS, 1× protease inhibitor complete EDTA-free cocktail, Roche). Nuclei were sonicated using a Bioruptor ultrasonic cell disruptor (Diagenode) to shear genomic DNA to an average fragment size of 200 bp. 20 μg of chromatin was diluted to a total volume of 500 μl with ChIP buffer (16.7 mM Tris-HCl, pH 8.1, 167 mM NaCl, 1.2 mM EDTA, 0.01% SDS, 1.1% Triton X-100) and incubated overnight with the ChIP-grade antibodies against H3K9me2, H3K9me3, H3K27me3, H3K4me1, H3K4me3 and H3K27ac. After washing, bound chromatin was eluted with the elution buffer (1% SDS, 100 mM NaHCO₃). Upon proteinase K digestion (50 °C for 3 h) and reversion of cross-linking (65 °C, overnight), DNA was purified with phenol/chloroform and ethanol precipitated.

The quality and quantity of the isolated DNA were determined with a Qubit® 4 Fluorometer (Life Technologies, California, USA) and the 4200 TapeStation system. Briefly, ChIP samples (1 ng) were end-repaired and polyadenylated before the ligation of Illumina-compatible adapters. The adapters contain the index for multiplexing. The quality and quantity of the enriched libraries were validated using Qubit® 4 Fluorometer and the 4200 TapeStation system. The libraries were pooled to 10 nM in Tris-Cl 10 mM, pH8.5, with 0.1% Tween 20. Sequencing was performed on the NovaSeq 6000 for 100 bp single-end reads (Illumina, Inc., California, USA).

For F123-ESCs, an N-masked genome was created using the mm10 reference genome and the Variant Call Format (VCF) file from the mouse genome project containing the SNP information for all strains. The SNPsplit package[58] was used to read the VCF file and extract the SNP information for the S129/SvJae and CAST strains and N-mask for

the mm10 genome. ChIPseq reads were aligned to the N-masked mm10 reference genome with Bowtie2 (version 2.5.0[54]) using default parameters. PCR duplicates were removed using samtools[55]. SNPsplit was then used to tag the SNPs for S129/SvJae and CAST alleles in the bam file and sort the haplotype-specific bam files for both alleles. Bigwig files were created from the bam files using bamCoverage from deeptools[57] using default parameters. Integrative Genome Viewer (IGV, version 2.15.4)[59] was used to visualize and extract representative ChIPseq tracks.

## Gene ontology (GO) enrichment

GO enrichment analysis of genes was performed with DAVID 6.8[67]. All genes in *Mus musculus* were used as the background. Data visualization was performed on Graphpad Prism (version 10.5.0).

## DNA−FISH

DNA−FISH probes for chromosomes 1, 2, 5, and 19 were generated as previously described[20]. The probes were obtained with oligopaint libraries that were constructed using the PaintSHOP interface created by the Beliveau lab and were ordered from CustomArray/Genscript in the 12 K Oligopool format. Each library contains a universal primer pair used to amplify all the probes in the library, followed by a specific primer pair hooked to the 40−46-mer genomic sequences, for a total probe of around 124−130-mers. Oligopaint libraries were produced by emulsion PCR from the pool, followed by a "two-step PCR" and lambda exonuclease as described before[20,68]. Specifically, the emulsion PCR with the universal primers allowed amplifying all pool of probes in the library. The "two-step PCR" led to the addition of a tail to the specific probes bound to the Alexa Fluor 488 fluorochrome. All oligonucleotides were purchased from Integrated DNA Technologies (IDT, Leuven, Belgium). All oligonucleotide sequences and the mm10 coordinates for the probe libraries are listed in Supplementary Data 4.

DNA-FISH was adapted from the protocol of Cavalli's lab[69]. Cells fixed with 4% PFA for 10 min at room temperature, 24 h after seeding. Cells were washed three times with 1× PBS and then permeabilized with 0.5%Triton-X100/PBS for 10 min on ice, followed by 10 min at room temperature. After three additional washes with 1× PBS, cells were incubated with 0.1 M HCl for 10 min at room temperature, washed twice with 2× SCCT, and twice with 50% Formamide-2× SCCT for 20 min, firstly at room temperature and then at 60 °C. Probe mixture contained around 45 pmol of Oligopaint probe, 0.8= μL of ribonuclease A (ThermoScientific, 10 mg/mL), and FISH hybridization buffer for a total mixture volume of 20 μL, which was directly added to the slide. Cell DNA was denatured at 83 °C for 8 min, and hybridization was performed in a humid dark chamber overnight at 42 °C. Cells were washed with 2× SCCT for 15 min once at 60 °C and once at room temperature, once with 0.2× SCC for 10 min, twice with 1× PBS for 2 min, and three times with 1× PBT for 2 min. Slides were then incubated for 1 h at room temperature with blocking solution in a dark, humid chamber, and overnight at 4 °C with primary antibody against Nucleolin (Abcam Cat# ab22758). Cells were washed four times with 1× PBT at increasing incubation times (1 × 2 min, 1 × 3 min, 2 × 5 min) and then incubated for 2 h at room temperature with goat anti-mouse IgG (H + L) Alexa Fluor 546 (ThermoFischer Cat# A11035) in a dark, humid chamber. Slides were washed four times with 1× PBT at increasing incubation times (1 × 2 min, 1 × 3 min, 2 × 5 min), three times with 1× PBS for 2 min. Then, slides were mounted with Vectashield DAPI mounting media (Vector, H-1200) and, after drying, stored at 4 °C.

DNA FISH/IF samples were imaged using a Zeiss LSM 980 Airyscan, with a z-stack collected for each channel (step size, 0.3 μm), using the oil objective Plan-Apochromat/1.40. Images were processed by ImageJ (version 2.14.0/1.54 F). The individual cells were identified by Hoechst/ DAPI staining, and cells containing a signal for the DNA−FISH channel were identified manually on the corresponding fluorescent channel. The distance between the DNA−FISH signal and the nucleolar marker immunofluorescence signal was calculated using ImageJ (version 2.14.0/ 1.54 F) and used to count the number of foci contacting the nucleolus and the number of cells with at least one contact with the nucleolus.

## Immunofluorescence

Cells were plated on coverslips (VWR, ECN 631–1577) and pre-coated with Matrigel (Corning, 356238) for 1 h at 37 °C. After 48–72 h, depending on the treatment, cells were fixed using 3.7% formaldehyde (Sigma-Aldrich, 47608) in PBS for 10 min at room temperature. After two washes in PBS, cells were permeabilized with 0.5% TritonX-100 (Sigma-Aldrich, T8787) in PBS for 10 min on ice and 10 min at room temperature. Cells were then washed two times in PBS and blocked with 1% Bovine Serum Albumin (Sigma-Aldrich, A2153) in PBS-0.1% Tween 20 (Tween 20: Merck, P9416) for 1 h at room temperature in a humidified chamber. The coverslips were then incubated with a 1:500 dilution of primary antibodies (NPM1, Sigma-Aldrich Cat# B0556; UBF, ThermoFischer Cat# H00007343-M02; Nucleolin, Abcam Cat# ab22758) resuspended in blocking solution overnight at 4 °C in a humidified chamber. After 3 washes in 1× PBS, the coverslips were incubated with 1:500 diluted secondary Alexa Fluor® antibody resuspended in Blocking Solution for 1 h at room temperature. After three washes in 1× PBS, the coverslips were mounted with VECTASHIELD® Antifade Mounting Medium with DAPI (H-1200-10), sealed with nail polish and imaged within one week.

Nucleolar volumes were quantified using Imaris (version 10.2) based on immunofluorescence staining of the nucleolar marker NPM1 (Sigma-Aldrich Cat# B0556), while nuclear volumes were defined using DAPI. The volume of nucleolar-associated chromatin was estimated by adding 0.6 μm to the measured nucleolar radius, corresponding to the average width of the perinucleolar shell marked by H3K9me2, which delineates NADs[31]. Based on this spatial definition, the nucleolar-associated volume accounted for ~65% of the total nuclear volume of the laser-microdissected samples. Using the cumulative distribution function for nucleolar contact frequency threshold of >20% in 53 microdissected ESCs, the false positive rate was estimated to be 0.000144%.

## Published datasets

Published datasets used in the study were from: H3K27ac-, H3K4me1-, H3K27me3-ChIP-seq[53] (GSE72164) [https://www.ncbi.nlm.nih.gov/geo/query/acc.cgi?acc=GSE72164]; H3K4me3- and H3K9me3-ChIPseq[70] (GSE23943) [https://www.ncbi.nlm.nih.gov/geo/query/acc.cgi?acc=GSE23943]; H3K9me2-ChIPseq[71] (GSE77420) [https://www.ncbi.nlm.nih.gov/geo/query/acc.cgi?acc= GSE77420]; A/B compartments[72] (GSE96107) [https://www.ncbi.nlm.nih.gov/geo/query/acc.cgi?acc=GSE96107]; Nucleolar DamID and RNAseq in ESCs and NPCs[20] (GSE150822) [https://www.ncbi.nlm.nih.gov/geo/query/acc.cgi?acc=GSE150822]; Lamin-DamID[73] (GSE17051) [https://www.ncbi.nlm.nih.gov/geo/query/acc.cgi?acc=GSE17051].

## Data availability

The data supporting the findings of this study are available from the corresponding authors upon request. The NoLMseq, RNAseq, ChIPseq, and scRNAseq data generated in this study have been deposited in NCBI's GEO database under accession code GSE269201. Source data for the figures and Supplementary Figs. are provided as a Source Data file. Source data are provided with this paper.

## Code availability

All custom scripts are deposited on GitHub (https://doi.org/10.5281/zenodo.17256205)[74].

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

## Acknowledgements

This work was supported by the ERC grant (ERC-AdG-787074-NucleolusChromatin), Swiss National Science Foundation (31003B-201268 and 320030-227818). K.W. was supported by an EMBO Postdoctoral Fellowship (ALTF 209-2021) and a postdoc grant of the University of Zurich. We thank Catherine Aquino, Lennart Opitz, and the Functional Genomic Center Zurich for the assistance in sequencing. We thank the Scientific Center for Optical and Electron Microscopy of ETH Zurich for support and assistance in LCM. We thank Ana Pombo for her technical advice on laser-microdissection and library preparation, and Bing Ren for the F123-ESC line.

## Author contributions

K.W. and J.K-R. established the NoLMseq. K.W. performed NoLMseq in all the described conditions, ChIPseq and RNAseq in F123-ESCs, and all data analyses of NADs. S.G. and M.R. performed DNA-FISH analyses. C.M. performed IF analyses, and P.R. measured microdissection volumes. K.W. and R.S. wrote the paper. R.S. conceived and supervised the project.

## Competing interests

The authors declare no competing interests.
