## [Transparent Peer Review file · Nature Communications]

Single-cell dynamics of genome-nucleolus interactions captured by nucleolar laser microdissection (NoLMseq)

Corresponding Author: Professor Raffaella Santoro

The peer review of this manuscript started at another nature portfolio title and mentions of previous journals have been redacted in the transferred files. The manuscript was transferred to NCOMMS on 4 August 25. The referees have agreed to have their comments shared.

Version 0: No Peer Review

Appeal 1/Version 1: No Peer Review

Version 2:

Reviewer #1

This manuscript describes a novel approach (NoLMseq) to identify the genomic regions that are proximal to nucleoli. The method is clever and conceptually simple. While several other mapping methods have been reported to map nucleolus-genome proximity/interactions, NoLMseq stands out because it is a single-cell method. Setting this up is an impressive accomplishment. The only single-cell method that comes close is seqFISH+, but this seems to have a current resolution of ~1Mb, compared to ~100kb for NoLMSeq.

Both the text and the figures are generally clear. That said, in my view substantially more evidence is necessary to document the quality of the data. See below.

Because this is a new mapping technology, the results are largely descriptive and not particularly surprising considering what the previous nucleolus-mapping methods had already revealed. Nevertheless, the data still uncover some interesting findings, such as changes upon ActD treatment, proximity of ribosomal protein genes, and potential mono/bi-allelic proximity to nucleoli.

Major points:

1. Figure 1b shows only one example of a laser capture "hole". Please provide a more extensive set of images to document how this was done. A potential limitation of the NoLMseq method would be that it may be only feasible in cells with one large nucleolus, while many cell types have multiple smaller nucleoli (including some cells shown in figure 1b, and in figure 6d). Could this create a bias in the current study? Or is it possible to capture multiple nucleoli from one cell?
2. The single-cell NAD patterns shown in figure 1c seem almost random, with the exception

of the higher density on a subset of chr19 . Similar for Extended Data figure 1b. This is very different from single-cell lamina interactions, which, albeit stochastic, show clearly recurrent domain patterns (see papers by Jop Kind). Likewise, it is difficult to tell whether the contact frequencies in Extended figure 2 are very different from random. The low signals of the X chromosome and a slight enrichment on chromosomes with rDNA are encouraging, but overall the patterns seem much noisier than previously reported by other methods - although these methods also may suffer from some limitations. I am missing a critical assessment of this. It is natural that single-cell data are noisy, but it is important that the reader gets a good sense of this. Below I suggest some scatterplots that could offer some more insights into the noise levels.

3. As mentioned by the authors, one possible concern with NoLMseq is that (parts of) chromosomes above and below the nucleolus may also be captured, thus contaminating the data. It would be useful to have a quantitative estimate of this contamination. How much of the total nuclear volume is above and below the nucleolus? This could be estimated by confocal or super-resolution microscopy. And what do the single-cell maps look like for a set of control captures (capturing a similar sized nuclear region outside the nucleolus)?

4. The overlap with SPRITE and nucleolar DamID (Figure 1h) seems improbably high. It is unclear how this analysis was done; how is overlap defined? Much better (and more transparent) would be to plot NoLMseq contact frequency NoLMseq versus nucleolus interaction scores according to the other two assays in conventional scatter plots (so continuous rather than binarized data). What do these plots look like, and what is the correlation coefficient?

5. I find the claim of 2 populations (Figure 3) not entirely compelling. Hierarchical clustering by definition yields two populations at the highest bifurcation, so even random data would generate two populations; it may be good to randomize/bootstrap the data 1,000 times and see if the mild enrichment of some (but not all) histone marks are true, or just random fluctuations. Furthermore, the two populations remain enigmatic: what is their biological meaning? What two different states of the cells may it reflect?

6. Section starting on line 212: "Detachment of NADs upon differentiation into NPCs results in gene activation" incorrectly claims a causal relationship. It is only a correlation. The final sentence of this section also incorrectly claims a causal relationship. Moreover, the differences in gene activity shown in Figure 4f are tiny (0.1 log₂ unit ~ 15%). I am puzzled about figure 4g: usually in such fish-tail distributions, genes with lower

expression are less reliable and hence show fewer significant changes. But in this plot there are many blue and red dots in this unreliable range. The Methods do not specify how this analysis was done. If it was not done so, I strongly recommend to use DESeq2 or similar state-of-the-art package to do this RNAseq analysis. Please also specify the number of independent biological replicates (in the figure legend), and confirm that the ESC and NPC RNAseq data were generated side-by-side.

7. The preferential association of ribosomal protein genes with nucleoli is very surprising, considering their very high transcriptional activity. Could this be an artefact? Are they perhaps speckle-associated, and does NoLMseq pick up some speckle-associated DNA? Were the RP genes also detected by any of the other nucleolus-mapping methods? It may be useful to highlight these genes in the scatterplots suggested under point 4.

8. The bi-versus-mono-allelic analysis 6c seems tricky to me. If the data are noisy (particularly due to a high rate of false-negatives), then obviously the chance of capturing bi-allelic interactions becomes small. One would at least expect that rDNA loci show a high bi-allelic frequency; is this indeed the case?

Minor points:

9. It is incorrect to state that NoLMseq maps nucleolus *interactions* (title), which would suggest molecular contacts. It maps spatial proximity.

10. The lower NAD coverage in NPCs could be due to higher noise levels in the data. Autocorrelation analysis may help to assess this. Were the ESC and NPC data generated in parallel?

11. Please comment on the success rate of the method. Which proportion of captured nucleoli passed the QC criteria of Extended Figure 1a? And of these, did they all yield useful sequencing information, or were some datasets discarded? How robust is the method from experiment to experiment?

12. p6: "The identified reads over the reference genome showed a bimodal distribution, indicating that loci are either in a "contact" or "no contact" state with nucleoli (Fig. 1c, Extended Data Fig. 1b)": Figure 1c seems to show binarized data, and from Extended Data Fig 1b it is not obvious that this is a biomodal distribution. Please add a figure panel

showing a histogram / density plot that directly demonstrates the bimodal distribution.

13. Figure 2fgh: it does not make a lot of sense to do all these pairwise t-tests between different bins (because so many comparisons were done, it would require multiple-testing correction). The binning also may obscure fine patterns in the overall relationships. Please replace by scatterplots that directly show the correlations between CF and RPKM, and provide correlation coefficients and associated p-values.

14. Figure 3b, right-hand panel: label at top should be "non-rDNA chromosomes"?

15. Are the genes that detach from the nucleolus in NPCs preferentially relocated to the lamina? This could be easily analysed using available DamID data (e.g. Peric Hupkes et al, Mol Cell 2013).

16. Figure 2a may be a case of circular reasoning? If the contact frequency is very high, then naturally the LAD coverage will be high too?

=====
Reviewed by Bas van Steensel, review task accepted: 19 Feb 2025;
completed 24 Feb 2025. It is my standard policy to sign and date *all* of my manuscript
review reports, regardless of my comments and recommendations. All correspondence
about this manuscript should go via the editor. PLEASE DO NOT REMOVE THIS NOTE
=====

Reviewer #2:

The authors employ a laser microdissection tool to isolate mouse ES nucleoli for DNA sequencing, enhancing the accuracy of NAD (NoLMseq) identification. This method enables single-cell resolution of nucleolar-associated DNA, providing a more precise representation of associated DNA domains. The study offers new insights into the role of nucleoli in genome organization, particularly the spatial association of ribosome synthesis-related genes with nucleoli. Furthermore, nucleolar stress disrupts NAD organization, correlating with changes in gene expression. The discovery of two distinct NAD populations—one more transcriptionally active than the other—in mES cells is particularly interesting. Overall, this study introduces a more accurate approach to analyzing nucleolar-associated genome domains and uncovers novel insights into their functional significance, making it suitable for publication in this journal. The following are specific points for consideration:

1) It would be insightful to include an analysis of NAD dynamics across the two identified

populations during the transition from ES cells to NPCs, providing a deeper understanding of nucleoli and NADs organization during differentiation.

2) The authors could further discuss the differences in cell cycle length and ribosome biogenesis between NPCs and ES cells examined in this study, particularly in relation to potential alterations in NAD organization.

3) An elaborated comparison of NADs identified in ES and NPC cells using the improved method with those reported in the authors' 2022 paper would strengthen the discussion.

Reviewer #3:

The genome organization around nucleoli and its functional implications remains a crucial question in nuclear architecture. Several methods, including sequencing of chromatin associated with purified nucleoli, nucleolar-DamID, and TSA-Seq, have been employed to map nucleolus-associated domains (NADs). In this study, Walavalkar et al. developed NoLMseq, a new method that employs laser-capture microdissection to directly isolate the chromatin near nucleoli for mapping NADs at single-cell resolution. This approach offers a unique advantage in that it does not rely on direct molecular interactions between nucleolar components and chromatin, making it well-suited to study NADs under stress conditions, where nucleolar composition and properties may change. The authors show that the NADs identified by NoLMseq are consistent with previous studies, but also reveal a significant degree of heterogeneity at the single-cell level. Furthermore, they demonstrate that NADs undergo dynamic changes during ES cell differentiation and upon nucleolar stress, with the detachment of genomic regions from nucleoli being associated with gene expression changes. Overall, these data suggest that NoLMseq is a valid method for mapping NADs and may provide information that is unattainable using other methods.

While the study is interesting from a methodological standpoint, my main concern is that it provides limited novel biological insights at the current stage. The observation that a large fraction of NADs detach from nucleoli during the differentiation of ESCs into NPCs or upon nucleolar disruption has been reported previously (Bersaglieri et al. 2022, Vertii et al. 2019). Although the authors correlate NAD detachment with gene activation through bulk RNA-Seq, the direct role of nucleolar association in gene regulation remains unproven. One of the key advantages of NoLMseq is its ability to resolve NAD heterogeneity at the single-cell level. The authors uncover several intriguing findings, such as the classification of ESCs into two populations based on their NAD organization and the observation that most NADs associate with nucleoli in a monoallelic manner. However, the functional implications of these findings are not explored. These and other suggestions are discussed

in greater detail below.

1. Lines 121-125: It is common for a single ES cell nucleus to harbor multiple nucleoli. Did the authors dissect only one nucleolus from each nucleus, or did they dissect all nucleoli from the same nucleus and combine them in one tube for later analysis? While this may not affect NAD identification, it could influence interpretations of NAD heterogeneity across cells. For instance, when the authors classify ESCs into two populations based on nucleolar contacts, could this be due to the capture of different nucleoli rather than true biological variability?
2. Lines 182-185: The authors found that the NAD/LAD (Type-2) regions exhibit overall lower CF values compared to NAD-only (Type-1) regions and conclude that “NAD/LAD domains rarely contact both nucleolus and NL in the same cell”. I find this logic unclear. It seems plausible that Type-2 regions represent a different type of heterochromatin than Type-1, and therefore associate with the nucleoli at a lower frequency.
3. The observation that ESCs can be classified into two populations, with chromosomes 12 and 19 preferentially associated with nucleoli in one group and chromosomes 16 and 18 in the other, is compelling. I suggest the authors conduct DNA-FISH to further validate this result.
4. Related to the last point, what are the functional implications of these distinct patterns of NAD organization in different ESCs? The authors should explore the relationship between NAD heterogeneity and gene expression by performing RNA-FISH. Additionally, the authors could analyze available scRNA-Seq data to determine whether gene expression from NADs in different ESCs can also be categorized into two populations.
5. Lines 226-228: The authors observe that genes within domains that detach from nucleoli upon ESC differentiation to NPCs (ESCsp-NADs) are more highly expressed in NPCs, leading them to conclude that detachment facilitates gene activation. However, the authors should include appropriate controls for the analysis in Figure 4f-i, such as genes within common NADs, NPCsp-NADs, and non-NADs within Compartment A, to support this conclusion. Ideally, RNA-FISH should also be performed to validate this finding.
6. While this study demonstrates that the ESCsp-NADs genes are activated in NPCs, a previous study from the same group (Bersaglieri et al. 2022) stated that the ESCsp-NADs genes do not exhibit significant gene expression changes and that the detachment from nucleoli unlocks these genes for activation in later stages. What is the reason for this discrepancy? The NoLMseq method likely revealed the real changes in NAD organization. However, the authors should discuss the differences between the two studies in more detail.
7. In Figures 5h and 5i, the authors show that genes within NADs that detach upon ActD treatment are enriched in upregulated genes versus downregulated genes. While this

finding is intriguing, I recommend that the authors compare the fraction of upregulated, downregulated, and unchanged genes across three groups: detached NADs, NADs that remain attached, and non-NAD regions. Statistical analyses, such as the Chi-square test, should be performed to help assess the significance of these observations.

8. Lines 277 to 279: The authors stated, “The detachment of genes from nucleoli, when nucleolar integrity is compromised, leads to their transcriptional activation”. I think this is an overstatement, as the authors have not conducted experiments to prove causality. Additionally, even within detached NADs, while 906 genes show upregulation, 705 genes are downregulated (Figure 5i). Thus, the effect of nucleolar association on gene expression appears to be facilitative and context-dependent. The authors should also exercise caution when making similar claims in the discussion.

9. The finding that most NADs associate with nucleoli in a monoallelic manner is also intriguing. As with my previous comment (point 4), I suggest the authors either perform RNA-FISH or analyze available scRNA-Seq data to investigate how the allelic association with nucleoli affects gene expression.

Appeal/Version 3: No Peer Review

Version 4:

Reviewer #1:

GENERAL:

The authors have addressed several of my concerns (see below; numbering as in the rebuttal). But my major points 1-4 need additional attention. Point 3 is a very difficult one: from the responses and new analyses, it seems to me that NoLM-seq has an intrinsic limitation that causes it to be quite noisy - it often captures chromosomal regions that are not nucleolus-associated, but happen to be above or below the nucleolus. This substantially dampens my enthusiasm for the technique. Perhaps the method will be only really useful when applied to very flat cells with "pancake" nuclei such as fibroblasts? At the very least, the authors should be very explicit about this limitation in the main text (instead of tucking it away in the Methods), so readers can decide for themselves. Points 2 and 4 are also relevant in this context.

Important general remark: all code for data processing and data analysis (ideally a separate script or markdown file for each figure) should be made available, preferably on Github. Currently this is missing. Combined with the very cursory description of the computational

analyses this is not acceptable. It seems misleading that the Reporting Summary states: "No software was used for data collection" - obviously a lot of computational work was needed for the processing and visualization of the data.

MAJOR POINTS:

1. The new figure 1b still shows only one nucleolus. This is not sufficient for giving the readers a good view of how the microdissections are done. Please add a set of ~ 10 representative images collected throughout this project.

Furthermore, the authors only address the issue of multiple nucleoli *in their rebuttal*, but do not discuss this *in the manuscript*: how often did the cells have multiple nucleoli for the collected data, and how was this issue handled (were all nucleoli, including very small ones, captured; or was only one big nucleolus captured?). Bottom line: precise documentation and transparency in the manuscript is needed.

Minor point: It is puzzling that the authors respond in their rebuttal: "It is certainly possible to cut two nucleoli, however, we did not include this in our analysis since the large majority of ESCs have one large nucleolus." Then why did they show in their rebuttal two images of nuclei in which multiple nucleoli are captured?

2. I appreciate the newly added random simulations. But there seems to be a flaw in this: I understand that the 53 random datasets all were taken to have 15% coverage. The real data however show a broad distribution of coverages (figure 1f). The random datasets should have the same distribution of coverages. I could not find a description of the randomization in the Methods section, so it is not clear how this was exactly done.

3. The new estimate that only 34% of the captured volume is nucleolar puts the results in a very different light. If two-thirds of the captured volume is above and below the nucleoli, and considering that nucleoli (one-third of the volume) contain very little DNA, then this method primarily captures non-nucleolar DNA! This reduces my enthusiasm for the method; most of the domains visible in figure 1c may be simply chunks of chromosomes that happen to be above or below the captured nucleolus. This explains why there seems so little concordance between individual nuclei, unlike Jop Kind's single-cell maps of lamina interactions.

Furthermore, proper documentation is needed of the 34%: the readers need a figure showing how the measurement was exactly done, plus a plot of the distribution of percentages across a set of ~50 nuclei.

4. I respectfully disagree. Binary analyses always require arbitrary cutoffs, which can greatly affect the outcome. Visualization of continuous data is much more informative. For the sake of transparency, scatterplots should really be shown. Of course the authors may then discuss their interpretation of potential caveats, but handwaving in a rebuttal only is not acceptable. Furthermore, the new scatterplot in Extended Fig 3b (comparing the results to SeqFISH, another method) shows the value of scatterplots: a clear forked distribution is visible, pointing to a systematic bias in at least one of the two methods. It seems to me that systematic analysis of scatterplots comparing NoLMseq to DamID, SPRITE and SeqFISH could help to find out which method deviates from the others.

Moreover, I still do not understand how the overlap in figure 1h was calculated. Please provide a description including formula in the Methods. Considering point 3 above, how can the overlap be close to 100%? This really does not make sense.

5. OK, thanks for adding these new analyses.

6. OK.

7. Thanks for the scatterplot, which boosts confidence in the ribosomal genes observation.

8. OK.

MINOR POINTS:

9. I still think that the word "interactions" in the title is inappropriate. It is odd that the authors agree with me that this word should be avoided, but that they keep it in the title. Proximity is better, particularly considering point 3 above.

10. I disagree that the ACFs are similar. Clearly, the autocorrelation is worse in NPCs (shifted substantially to the left!), and hence the data are probably more noisy. I do not understand how the Pearson correlation was calculated, but it seems not the right measure to determine the shift in ACF.

11. OK, but please document this in the manuscript, ideally with small table or figure, not just in a response in the rebuttal.

12. OK.

13. The new scatterplots shed new light on the links with H3K27ac and H3K9me2. The correlations turn out to be extremely weak! Correlation coefficients are not provided, but by eye they seem <0.2 . This should be discussed explicitly, and I think it is somewhat misleading to have these scatterplots tucked away in the Extended data.

14. OK.

15. OK.

16. OK.

=====Reviewed by Bas van Steensel, review task received: 12 May 2025; completed 24 May 2025. It is my standard policy to sign and date *all* of my manuscript review reports, regardless of my comments and recommendations. All correspondence about this manuscript should go via the editor. PLEASE DO NOT REMOVE THIS NOTE
=====

Reviewer #2:

The revised manuscript has addressed the reviewer's comments adequately. It is suitable for publication.

Reviewer #3:

In this revised manuscript, Walavalkar et al. have incorporated new experiments and data analyses that clarify the physiological implications of the NAD organization revealed by NoLMseq. In particular, by performing scRNA-Seq in ESCs, the authors show that distinct nucleolar-associated domain (NAD) organization patterns in two ESC populations may correlate with the expression of genes within different NADs. Moreover, the authors performed additional analyses to further support the conclusion that dissociation of NADs from the nucleolus during ESC-to-NPC differentiation, as well as upon Actinomycin D treatment, is associated with derepression of genes. The authors have also made textual edits that improve clarity and corrected some overstatements.

Overall, I find that these revisions have significantly strengthened the manuscript. Most of my earlier concerns have been addressed; however, I do have a few remaining points that require further attention by the authors.

Major points:

1. I appreciate the authors' efforts to integrate scRNA-Seq with NAD mapping to illustrate

that distinct nucleolar contact patterns in ESCs at the single-cell level may relate to differential gene expression and cellular states. By examining the expression of genes within NADs specific to the two ESC populations, the authors show that genes in population 1 NADs tend to co-express with pluripotency markers Nanog and Sox2, while those in population 2 NADs do not. This is an interesting observation. However, the current analysis is based on only five genes per NAD group. I suggest that the authors perform more comprehensive statistical analyses across a broader set of genes with detectable expression in the scRNA-Seq dataset to rigorously assess whether genes within the same NAD group show higher expression correlation compared to genes from different NAD groups.

Minor points:

2. Regarding lines 187 and 193 and my previous point #2: While I agree with the authors that it is plausible NAD/LAD regions may interact with the nucleolus in one cell and the nuclear lamina (NL) in another, the low CF values alone do not provide direct evidence for or against simultaneous nucleolar and NL contacts within the same nucleus. For instance, one could imagine a scenario where a NAD/LAD region contacts the nucleolus in 15% of cells, and in all such cases, it also contacts the NL. To avoid overinterpretation, I recommend removing the sentence in lines 186–188: “Since this classification of NADs was obtained from ESC population, it has remained elusive whether NAD/LAD regions contact both nucleoli as well as the NL in the same or different cells.”
3. In line 224, the authors show that by combining the more quantitative NoLMseq data and more stringent criteria, they were able to more accurately define the ESCsp-NADs and reveal the correlation between NAD changes and gene derepression. I suggest the authors include a panel in the extended data figure to illustrate the differences between ESCsp-NADs identified by NoLMseq and those previously defined using nucleolar DamID.
4. Lines 225–227: It would be helpful to mention here that genes located in NPCsp-NADs, common NADs, or in the A compartment do not exhibit significant expression changes.
5. Line 278: The authors suggest that ESC populations with distinct nucleolar contact patterns may represent “different developmental potentials.” However, these populations could simply reflect dynamic, fluctuating states of ESCs under in vitro culture conditions. I recommend softening this statement to avoid overinterpretation.

Single-cell dynamics of genome-nucleolus interactions captured by nucleolar laser microdissection (NoLMseq)

Walavalkar et al.

Response to the Reviewers

We thank all the Reviewer's for the positive and constructive comments. In this revised manuscript, we have provided additional data addressing all the Reviewer's requests and modified the text to clarify points that were not sufficiently clear. Furthermore, to more clearly differentiate the validation results of NoLMseq from the novel discoveries revealed by this method, we have reorganized the figures order by swapping Figures 3 and 4 and modified the text accordingly. The additional data strengthen the conclusions of our work.

List of changes

Modifications in the text were highlighted in red.

The new data are listed here below.

	Revised manuscript	Previous version	Description	Reviewer
1	Fig. 1b	Fig. 1b	New images of nucleolus microdissection	Rev. 1
2	Figs. 2f,g,h	Figs. 2f,g,h	New statistical analysis	Rev. 1
3	Fig. 4e	---	UMAP scRNAseq	Rev. 3
4	Fig. 4d	---	UMAP scRNAseq of NAD-genes in populations 1 & 2	Rev. 3
5	Fig. 5i	Fig. 5i	Quantification and statistical analysis of up- and downregulated genes at NADs and non-NADs in ESC+ActD.	Rev. 3
6	Ext. Data Fig. 1c	---	Analysis of randomized sequences for 53 nucleoli	Rev. 1
7	Ext. Data Fig. 2	Ext. Data Fig. 2	Inclusion of CF of randomized sequences for 53 nucleoli	Rev. 1
8	Ext. Data Fig. 3b	---	Scatterplot of SeqFISH+ and NoLMseq CF with ribosomal protein genes	Rev. 1
9	Ext. Data Fig. 3c-e	---	Scatterplot of histone PTM and gene expression levels vs nucleolar CF	Rev. 1
10	Ext. Data Fig. 4a	---	Autocorrelation analysis of ESCs and NPCs	Rev. 1
11	Ext. Data Fig. 4b	---	Comparisons of gene expression levels between ESC _{sp} , NPC _{sp} , and common NADs between ESCs and NPCs	Rev. 3
12	Ext. Data Fig. 5a	---	P -values after bootstrap randomization of NADs	Rev. 1
13	Ext. Data Fig. 5c	---	Quantification of gene expression in scRNAseq	Rev. 3
14	Ext. Data Fig. 5d	---	UMAP scRNAseq of Sox2	Rev. 3

15	Ext. Data Fig. 7c	---	Proportion of bi-allelic NADs at chr. with and without rDNA	Rev. 1
16	Suppl. Table 2	---	GO and pathways NAD-genes in in populations 1 & 2	Rev. 1,3
17	Rebuttal letter	---	Images of nucleolus microdissection	Rev. 1
18	Rebuttal letter	---	Autocorrelation decay patterns using randomized ESC and NPC datasets	Rev. 1
19	Rebuttal letter	---	IF images of ESCs	Rev. 3
20	Rebuttal letter	---	Heatmap of average expression of scRNAseq data	Rev. 3

Reviewer #1 (Remarks to the Author): This manuscript describes a novel approach (NoLMseq) to identify the genomic regions that are proximal to nucleoli. The method is clever and conceptually simple. While several other mapping methods have been reported to map nucleolus-genome proximity/interactions, NoLMseq stands out because it is a single-cell method. Setting this up is an impressive accomplishment. The only single-cell method that comes close is seqFISH+, but this seems to have a current resolution of ~1Mb, compared to ~100kb for NoLMseq. Both the text and the figures are generally clear. That said, in my view substantially more evidence is necessary to document the quality of the data. See below. Because this is a new mapping technology, the results are largely descriptive and not particularly surprising considering what the previous nucleolus-mapping methods had already revealed. Nevertheless, the data still uncover some interesting findings, such as changes upon ActD treatment, proximity of ribosomal protein genes, and potential mono/bi-allelic proximity to nucleoli.

Author. We thank the Reviewer for the appreciation of our study and the constructive input for the computational and statistical analyses of our data. Below are clarifications and additional analyses that demonstrate the specificity and quality of our data.

We would like to clarify that the '*non particularly surprising*' results presented in the initial part of this work served for the validation of this first single cell/nucleolus method to identify NADs, as recognized by the Reviewer. As explicitly stated in the manuscript, in the initial part of this work, we used established knowledge to validate NoLMseq. This includes the expected frequent association of rDNA chromosomes with nucleoli, the known nucleolar association of centromeric proximal sequences, the known repressive chromatin signature of NADs, and the known NAD reorganization during ESC differentiation into NPCs. These results are not a limitation of this study. Instead, they demonstrate the validity of NoLMseq. The surprising and "*interesting findings*" are shown after the validation part of this manuscript. These results include (1) the presence of ribosomal protein genes at nucleoli, (2) genome remodelling upon ActD treatment, (3) the mono-allelic features of NADs, (4) the presence of two ESC populations with distinct NAD profiles and chromatin states. These important results could only be obtained by a single cell technology like the NoLMseq and could not be captured by any other existing technologies. To clearly separate the validation experiments from the exploratory application of NoLMseq, we reorganized the order of the figures and modified the text accordingly.

Major points:

1. Figure 1b shows only one example of a laser capture "hole". Please provide a more extensive set of images to document how this was done. A potential limitation of the NoLMseq method would be that it may be only feasible in cells with one large nucleolus, while many cell types have multiple smaller nucleoli (including some cells shown in figure 1b, and in figure 6d). Could this create a bias in the current study? Or is it possible to capture multiple nucleoli from one cell?

Author. Here below are multiple images documenting how nucleoli were microdissected. These images reflect the standard approach we described in detail in the Methods section and serve to visually confirm the accuracy and reproducibility of our microdissection procedure. We have now replaced the original image in **Fig. 1b** with a representative example obtained during the course of this study, as the original image was generated several years ago at the very beginning of the project. However, we did not add all these images in the manuscript since we do not think they will add further information to the method section.

Yes, the limitation of NoLMseq is that it can be applied only to cells with one or few nucleoli like mESCs. However, there are many other biomedical relevant cell types that have this feature and suitable for use with NoLMseq such as hematopoietic stem cells, adult stem cell, T cells, some cancer cell types to name few. We clarified this point in the discussion. On the other side, NoLMseq offers several and unique advantages over other methods, as it is the only method likely to be applicable under conditions of nucleolar stress, as demonstrated by the analysis of ESCs treated with ActD (**Fig. 5**). Moreover, as stated by the Reviewer, NoLMseq is so far the only single cell method with 100 kb resolution to study nucleoli. It is certainly possible to

cut two nucleoli, however, we did not include this in our analysis since the large majority of ESCs have one large nucleolus.

2. The single-cell NAD patterns shown in figure 1c seem almost random, with the exception of the higher density on a subset of chr19 . Similar for Extended Data figure 1b. This is very different from single-cell lamina interactions, which, albeit stochastic, show clearly recurrent domain patterns (see papers by Jop Kind). Likewise, it is difficult to tell whether the contact frequencies in Extended figure 2 are very different from random. The low signals of the X chromosome and a slight enrichment on chromosomes with rDNA are encouraging, but overall the patterns seem much noisier than previously reported by other methods - although these methods also may suffer from some limitations. I am missing a critical assessment of this. It is natural that single-cell data are noisy, but it is important that the reader gets a good sense of this. Below I suggest some scatterplots that could offer some more insights into the noise levels.

Author. To assess whether the observed patterns arise from noise, we generated 53 “random samples” ensuring that each had genome coverage comparable to our single-nucleolus data (15%, **Fig. 1f**). In these random datasets, less than 0.5% of the genome exhibited contact frequencies above 30%, whereas in ESC NADs more than 10% of the genome exceeded this (**new Extended Data Fig. 1c**). Additionally, in the random datasets, there is almost no genomic region at 0% contact frequency, whereas in our single-nucleolus data, nearly 10% of the genome was absent from all 53 single-nucleolus datasets. This suggests that specific regions are preferentially associated with or excluded from nucleolar interactions, rather than the data being dominated by noise. Furthermore, when analysing the distribution of rDNA-chromosomes, non-rDNA chromosomes, and sex chromosomes in the random datasets, we found that their proportions remained similar across all contact frequencies, and as expected reflect their genomic sizes (**new Extended Data Fig. 1c**). This is in stark contrast to our experimental data, where rDNA-chromosome proportions increase with contact frequency, while proportions of X and Y chromosome decrease (**Fig. 1d**). Finally, in the **new Extended Data Fig. 2**, we included the contact frequencies over the chromosome from the random samples. If the experimental data were entirely random, we would expect relatively flat curves as seen when we plot contact frequencies over chromosome lengths from random samples. Instead, our results reveal clear enrichments and depletions, reinforcing the notion that the observed patterns are non-random. Finally, it is important to clarify that these NAD profiles were validated using orthogonal single cell approaches, such the comparison of CF between high-throughput DNA-FISH analyses (seqFISH+) and NoLMseq CF which show a high correlation (**Fig. 1i**). Taken together, these analyses indicate that our single-cell NAD data, while inherently stochastic, exhibit structured and reproducible patterns that distinguish them from random noise.

3. As mentioned by the authors, one possible concern with NoLMseq is that (parts of) chromosomes above and below the nucleolus may also be captured, thus contaminating the data. It would be useful to have a

quantitative estimate of this contamination. How much of the total nuclear volume is above and below the nucleolus? This could be estimated by confocal or super-resolution microscopy. And what do the single-cell maps look like for a set of control captures (capturing a similar sized nuclear region outside the nucleolus)?

Author. We performed confocal microscopy followed by 3D-reconstruction, which revealed that, on average, 66% of the microdissected cylindrical volume corresponds to the non-nucleolar region (i.e., nucleus + cytoplasm). This information has been included in the Method section. Importantly, since NADs typically surround the nucleolus, it is expected that a portion of this non-nucleolar space contains NADs. We think that the sequencing of control captures will not be informative as these would have random DNA but would still have some NADs due to their monoallelic nature. Moreover, since the nucleolus is mostly central in ESCs and ESCs are spherical, a similar sized 2D cut will lead to the capture of lesser genomic material and adjusting for it further complicates any realistic comparison to the nucleolar microdissection. Instead, simulated random collection of similar genomic contents as the nucleolus as described above addresses these concerns better.

4. The overlap with SPRITE and nucleolar DamID (Figure 1h) seems improbably high. It is unclear how this analysis was done; how is overlap defined? Much better (and more transparent) would be to plot NoLMseq contact frequency NoLMseq versus nucleolus interaction scores according to the other two assays in conventional scatter plots (so continuous rather than binarized data). What do these plots look like, and what is the correlation coefficient?

Author. We apologize to not have clearly explained this data. We calculated the proportion of NADs identified by NoLMseq that were defined as NADs by Nucleolar-DamID (Bersaglieri et al. 2022) or genomic domains interacting with rRNA as defined by SPRITE (Quidonoz et al., 2018). We modified the text to increase clarity.

We believe that plotting NoLMseq CF values together with Nucleolar-DamID scores from cell populations is not informative. The reason is that the score value in Nucleolar-DamID is the enrichment of Dam methylation in ESCs expressing the nucleolar histone H2B fused to Dam relative to ESCs expressing H2B-Dam. Thus, both Dam-fused proteins are very similar to each other since they are histones (i.e., they both bind DNA, but H2B-Dam-NoLS binds more DNA at nucleoli than H2B-Dam), explaining why the score values of the identified NADs, while statistically significant and reproducible, are not as high and are generally similar to each other. We do not doubt this DamID method since the identification of NADs has been validated with several orthogonal approaches like multiple DNA-FISH analyses and HiC-rDNA. Thus, Nucleolar-DamID is more a qualitative method as it cannot clearly determine if a NAD sequence is more frequently located at the nucleolus than another one as in the case of NoLMseq. This is different from the LAD scores since in LaminB1-DamID in cell population values are calculated relative to control cells expressing GFP-Dam, a protein completely different from LMNB1, which has also a certain preference for active chromatin. Finally, the data of **Fig. 1h**, showing that NADs identified by NoLMseq are similar to the NADs identified by previous methods in cell populations, is not a major result of this work as it represents one of the many validations presented in the manuscript (**Figs. 1-3**).

5. I find the claim of 2 populations (Figure 3) not entirely compelling. Hierarchical clustering by definition yields two populations at the highest bifurcation, so even random data would generate two populations; it may be good to randomize/bootstrap the data 1,000 times and see if the mild enrichment of some (but not all) histone marks are true, or just random fluctuations. Furthermore, the two populations remain enigmatic: what is their biological meaning? What two different states of the cells may it reflect?

Author. We want to thank the Reviewer for this important suggestion. As requested, to assess whether the differences in histone modifications arise from random fluctuations, we conducted a bootstrap analysis with 1,000 random permutations of the NADs of populations 1 and 2 (**Fig. 3d**). We then quantified how many times the two populations obtained from these bootstrap randomisations showed similar observed

enrichments. In none of these iterations did we observe enrichment values for H3K9me2, H3K4me1, or H3K27ac that matched or exceeded those observed in the actual data, supporting the specificity of these enrichments. These results strongly support that the observed enrichments for H3K9me2, H3K4me1, and H3K27ac in the two populations do not result from random fluctuations. We have discussed these data in the revised manuscript and included empirical and observed p-values in the **new Extended data Fig. 5a**. We performed GO and pathway analyses showing that the specific NAD-genes of population 2 have significant enrichments in pathways linked to development and pluripotency whereas NAD-genes of population 1 are linked to metabolism and signalling pathways (**new Supplementary Table 2**). Thus, the presence of these two major populations could reflect the well-documented heterogeneity of ESCs cultured in serum, the conditions used in this study. ESCs are characterized by high or low expression of the pluripotency marker Nanog, which reflect distinct developmental potentials. To test this, we performed scRNAseq. Nanog-high and Nanog-low ESCs can be visualized in the UMAP of the **new Fig. 4e**. We then examined the top 5 genes with the highest nucleolar interactions with specifically one of the two populations (i.e., genes of population 1 that have CF >30% in population 1 and <5% CF in population 2 and vice versa for genes of population 2). We found that 4 out of 5 genes of population 2 show a similar expression profiles of Nanog-high cells and Sox2 positive ESCs while only 1 gene of population 1 shows such a pattern (**new Fig. 4f, Extended data Fig. 5c,d**). Thus, populations 1 and 2 might represent populations with distinct developmental potential. Thus, the data revealed structural heterogeneity in the ESC genome, resulting in two major populations distinguished by their genome organization around nucleoli, each characterized by distinct chromatin states and eventually different developmental potentials.

6. Section starting on line 212: "Detachment of NADs upon differentiation into NPCs results in gene activation" incorrectly claims a causal relationship. It is only a correlation. The final sentence of this section also incorrectly claims a causal relationship. Moreover, the differences in gene activity shown in Figure 4f are tiny (0.1 log₂ unit ~ 15%). I am puzzled about figure 4g: usually in such fish-tail distributions, genes with lower expression are less reliable and hence show fewer significant changes. But in this plot there are many blue and red dots in this unreliable range. The Methods do not specify how this analysis was done. If it was not done so, I strongly recommend to use DESeq2 or similar state-of-the-art package to do this RNAseq analysis. Please also specify the number of independent biological replicates (in the figure legend), and confirm that the ESC and NPC RNAseq data were generated side-by-side.

Author. We revised the text to eliminate any unintended implication of a causal relationship.

As stated in the paragraph "Published datasets", the RNAseq of ESCs vs NPC was from our previous work (PMID: 35304483, GSE150822). We made this point clear in Result section. We confirm that we used DESeq2, each condition was in triplicate, and that ESC and NPC RNAseq data were generated side-by-side.

Fig. 5f showed the \log_{10} (RPKM+1) for all the genes that are at NADs specifically in ESCs but not in NPCs (ESCsp-NAD genes, up, down and unchanged). This means that the difference between all genes is >50%. The ESCsp-NAD genes significantly affected are shown in **Fig. 4g,h**.

7. The preferential association of ribosomal protein genes with nucleoli is very surprising, considering their very high transcriptional activity. Could this be an artefact? Are they perhaps speckle-associated, and does NoLMseq pick up some speckle-associated DNA? Were the RP genes also detected by any of the other nucleolus-mapping methods? It may be useful to highlight these genes in the scatterplots suggested under point 4.

Author. As stated in the Discussion section (now lanes 398-399), our data are consistent with previous studies based on imaging analyses in living or fixed yeast cells showing the proximity of ribosomal protein genes and tRNA genes to nucleoli. As requested, we re-analysed the published Nucleolar-DamID and found that 31% of ribosomal protein genes are at NADs and included this information in the revised manuscript. Moreover, we have included a scatter plot showing that the CFs of ribosomal protein genes with nucleoli

from seqFISH+ data highly correlate with the corresponding NoLMseq CF values (**new Extended Data Fig. 3b**). Taken together, these data allow us to confidently exclude the possibility that the nucleolar association of ribosomal protein genes is an artifact.

8. The bi-versus-mono-allelic analysis 6c seems tricky to me. If the data are noisy (particularly due to a high rate of false-negatives), then obviously the chance of capturing bi-allelic interactions becomes small. One would at least expect that rDNA loci show a high bi-allelic frequency; is this indeed the case?

Author. We analysed the frequency of bi-allelic NADs on rDNA- and non-rDNA chromosomes and found that rDNA-chromosomes have about 10-fold higher proportion of biallelic NADs compared to non-rDNA chromosomes. We included this information in the revised manuscript (new **Extended Data Fig. 7c**). We cannot analyse the rDNA loci since these are repetitive sequences and were not mapped for the SNVs. We would like to strength that several data supported the mono-allelicity of NADs like DNA-FISH (**Fig. 6d**) and the higher CF of biallelic NADs compared to monoallelic NADs (**Extended data Fig. 7d**). Furthermore, analyses of histone modifications and gene expression between mono- and bi-allelic NADs indicate that bi-allelic NADs are in a more repressed chromatin and gene expression states (**Figs. 6f,g**). As described in the responses above, the false-negative rates do not seem to be leading to such specific differences between mono- and biallelic NADs. These findings collectively support our observations and argue against noise as the primary driver of bi- versus mono-allelic NAD patterns.

Minor points:

9. It is incorrect to state that NoLMseq maps nucleolus *interactions* (title), which would suggest molecular contacts. It maps spatial proximity.

Author. We agree. Accordingly, the term “interactions” was not used in the rest of the manuscript. However, we think that for a title the term “interactions” would be much clearer for a broad readership as done in the past for many LADs papers.

10. The lower NAD coverage in NPCs could be due to higher noise levels in the data. Autocorrelation analysis may help to assess this. Were the ESC and NPC data generated in parallel?

Author. The reduction of contacts with nucleoli in NPCs compared to ESCs is consistent with previous work using Nucleolar-DamID and rDNA-HiC (PMID: 35304483).

We performed the autocorrelation analysis as suggested by the Reviewer. The autocorrelation and decay between ESCs and NPCs seem very comparable, suggesting similar levels of signal over noise ratios (SNRs). We included this data in the revised manuscript (**new Extended data Fig. 4a**).

To assess whether the observed autocorrelation decay patterns were distinct from random noise, we performed the same analysis using randomized ESC and NPC datasets while preserving their respective genome coverage in single cells (see image below). As expected, the randomized samples exhibited a completely random decay pattern, confirming that the autocorrelation observed in the actual data reflects biologically relevant nucleolar associations rather than noise.

NPCs were generated in parallel to ESCs but were dissected and thus processed at different times. This is also true for samples themselves. No batch effects were seen from any of the sequencing runs.

11. Please comment on the success rate of the method. Which proportion of captured nucleoli passed the QC criteria of Extended Figure 1a? And of these, did they all yield useful sequencing information, or were some datasets discarded? How robust is the method from experiment to experiment?

Author. For ESCs, the success rate of the captured nucleoli to pass the QC criteria was 95% and of these about 90% yielded useful sequencing information. The method is very robust among different experiments. Samples with little to no alignment to the mm10 genome after sequencing were discarded.

12. p6: "The identified reads over the reference genome showed a bimodal distribution, indicating that loci are either in a "contact" or "no contact" state with nucleoli (Fig. 1c, Extended Data Fig. 1b)": Figure 1c seems to show binarized data, and from Extended Data Fig 1b it is not obvious that this is a bimodal distribution. Please add a figure panel showing a histogram / density plot that directly demonstrates the bimodal distribution.

Author. We apologise for the confusion. The term "bimodal distribution" was not used as a statistical meaning. It was a way to describe that the data show that loci are either in a "contact" or "no contact" state with nucleoli. We modified the text to avoid confusion.

In **Fig. 1b**, each black bar denotes that the region contacts the nucleolus and the white region indicates no contact. We clarified this in the corresponding figure legend. **Extended Data Fig. 1b** shows how the data look like after alignment to the mm10 genome. There are some regions that contain reads and some other regions, and sometime an entire chromosome, that do not.

13. Figure 2fgh: it does not make a lot of sense to do all these pairwise t-tests between different bins (because so many comparisons were done, it would require multiple-testing correction). The binning also may obscure fine patterns in the overall relationships. Please replace by scatterplots that directly show the correlations between CF and RPKM, and provide correlation coefficients and associated p-values.

Author. According to the request of the Reviewer, we calculated again the statistical significance (Q-values) using multiple-testing correction by the Benjamini and Hochberg method. The results remain the same. We have included these data in the revised manuscript.

We have also provided scatterplots. However, we decided to place them in the **Extended data Figs. 3c-e** since they are less informative than the boxplots showed in **Figs. 2f-h**. Indeed, the boxplot representation of the data allows comparisons with genomic domains within the A compartment, the NAD-only, and NAD/LAD regions.

14. Figure 3b, right-hand panel: label at top should be "non-rDNA chromosomes"?

Author. Thank you! We corrected it.

15. Are the genes that detach from the nucleolus in NPCs preferentially relocated to the lamina? This could be easily analysed using available DamID data (e.g. Peric Hupkes et al, Mol Cell 2013).

Author. We performed this analysis using the LAD profile of Peric Hupkes et al. We found that 21.56% of the NADs and 11.83% of the genes that detach from nucleoli relocate to the lamina in NPC. Notably, a large fraction of these genes (95%) do not show changes in gene expression between ESC and NPCs. We did not include these data in the revised manuscript, as the analysis of NADs and LADs during ESC differentiation into NPCs was extensively addressed in our previous work using Nucleolar-DamID. In this study, this analysis served primarily to validate the application of NoLMseq in a cell type other than ESCs.

16. Figure 2a may be a case of circular reasoning? If the contact frequency is very high, then naturally the LAD coverage will be high too?

Author. We think the Reviewer means **Fig. 2e**.

The data in simply showed that NAD-only regions are more frequently found in microdissected nucleoli than NAD/LAD regions. This makes sense, as NAD-only regions do not contact the nuclear lamina and therefore can be found more often in nucleoli, whereas NAD/LAD regions can also be located at the nuclear lamina. The important point here is that these measurements could only be achieved by single cell analyses. Moreover, these results further confirm that NoLMseq accurately identified NADs. We clarified this point in the manuscript.

=====Reviewed by Bas van Steensel, review task accepted: 19 Feb 2025; completed 24 Feb 2025. It is my standard policy to sign and date *all* of my manuscript review reports, regardless of my comments and recommendations. All correspondence about this manuscript should go via the editor. PLEASE DO NOT REMOVE THIS NOTE =====

Reviewer #2 (Remarks to the Author):

The authors employ a laser microdissection tool to isolate mouse ES nucleoli for DNA sequencing, enhancing the accuracy of NAD (NoLMseq) identification. This method enables single-cell resolution of nucleolar-associated DNA, providing a more precise representation of associated DNA domains. The study offers new insights into the role of nucleoli in genome organization, particularly the spatial association of ribosome synthesis-related genes with nucleoli. Furthermore, nucleolar stress disrupts NAD organization, correlating with changes in gene expression. The discovery of two distinct NAD populations—one more transcriptionally active than the other—in mES cells is particularly interesting. Overall, this study introduces a more accurate approach to analyzing nucleolar-associated genome domains and uncovers novel insights into their functional significance, making it suitable for publication in this journal.

Author. We thank the Reviewer for the appreciation of our study.

The following are specific points for consideration:

1) It would be insightful to include an analysis of NAD dynamics across the two identified populations during the transition from ES cells to NPCs, providing a deeper understanding of nucleoli and NADs organization during differentiation.

Author. This is a very interesting question; however, we believe that NoLMseq is not suitable for this analysis. The differentiation of ESCs into NPCs was performed using a previously established protocol (PMID:15332090) that yields a homogeneous Pax-6-positive NPC population after 8 days. However, the process itself involves a transiently heterogeneous cell population, with some cells differentiating faster than others. NoLMseq is well suited for homogeneous cell populations with few or one single nucleoli, such as ESCs and NPCs. However, its application to heterogeneous populations would be extremely challenging, as it would also require a substantial increase in the number of microdissected nucleoli (well over 50 for each time points) to distinguish subpopulations. We think the point raised by the Reviewer is very important and we are currently developing methods to analyse NADs in heterogeneous biological populations.

2) The authors could further discuss the differences in cell cycle length and ribosome biogenesis between NPCs and ES cells examined in this study, particularly in relation to potential alterations in NAD organization.

Author. These are very complex topics. To our knowledge, there are no comprehensive studies directly comparing the cell cycle dynamics between ESCs and NPCs in the context of nucleolar organization. With regard to ribosome biogenesis, it is very well known that ESCs display high levels of rRNA transcription and elevated ribosome biogenesis compared to differentiated cells, which also accounts for their characteristically large nucleoli (Reviewed in PMID: 32976768). However, it remains unclear whether a hyperactive nucleolus is responsible for changes in NADs between ESCs and NPCs. This is primarily because the molecular mechanisms underlying the anchoring of chromatin to the nucleolus are still not fully understood. As discussed in the Result section, we believe that these differences mainly stem from changes in global genome organization and chromatin composition between ESCs and NPCs, with ESCs harbouring a more open and dynamic chromatin than differentiated cells.

3) A elaborated comparison of NADs identified in ES and NPC cells using the improved method with those reported in the authors' 2022 paper would strengthen the discussion.

Author. We thank the Reviewer for this valuable suggestion.

The NADs identified in ESCs and NPCs using NoLMseq provide a more quantitative assessment of nucleolar associations compared to our previous approach using Nucleolar-DamID (Bersaglieri et al., 2022). While nucleolar DamID allows to classify genomic regions broadly as associated or not associated with the nucleolus, and these data were validated by multiple DNA-FISH analyses and rDNA-HiC, it lacked

the sensitivity to quantify the degree of association. In contrast, NoLMseq enables the quantification of nucleolar CF and the use of stringent criteria to robustly define NADs, as in the case of ESC_{sp}-NADs that were defined as high-confidence NADs in ESCs that were consistently excluded from the nucleolus (>25% CF in ESCs, <10% CF in NPCs). Thus, this quantitative resolution provides a more nuanced view of nucleolar genome organization and enabled us to detect changes that were only partially captured with Nucleolar-DamID. We have included this information in the Discussion of the revised manuscript.

Reviewer #3 (Remarks to the Author):

The genome organization around nucleoli and its functional implications remains a crucial question in nuclear architecture. Several methods, including sequencing of chromatin associated with purified nucleoli, nucleolar-DamID, and TSA-Seq, have been employed to map nucleolus-associated domains (NADs). In this study, Walavalkar et al. developed NoLMseq, a new method that employs laser-capture microdissection to directly isolate the chromatin near nucleoli for mapping NADs at single-cell resolution. This approach offers a unique advantage in that it does not rely on direct molecular interactions between nucleolar components and chromatin, making it well-suited to study NADs under stress conditions, where nucleolar composition and properties may change. The authors show that the NADs identified by NoLMseq are consistent with previous studies, but also reveal a significant degree of heterogeneity at the single-cell level. Furthermore, they demonstrate that NADs undergo dynamic changes during ES cell differentiation and upon nucleolar stress, with the detachment of genomic regions from nucleoli being associated with gene expression changes. Overall, these data suggest that NoLMseq is a valid method for mapping NADs and may provide information that is unattainable using other methods. While the study is interesting from a methodological standpoint, my main concern is that it provides limited novel biological insights at the current stage. The observation that a large fraction of NADs detach from nucleoli during the differentiation of ESCs into NPCs or upon nucleolar disruption has been reported previously (Bersaglieri et al. 2022, Vertii et al. 2019). Although the authors correlate NAD detachment with gene activation through bulk RNA-Seq, the direct role of nucleolar association in gene regulation remains unproven. One of the key advantages of NoLMseq is its ability to resolve NAD heterogeneity at the single-cell level. The authors uncover several intriguing findings, such as the classification of ESCs into two populations based on their NAD organization and the observation that most NADs associate with nucleoli in a monoallelic manner. However, the functional implications of these findings are not explored. These and other suggestions are discussed in greater detail below.

Author. We thank the Reviewer for the appreciation of our method.

Regarding the “*limited novel biological insights*” of our study and, conversely, the uncovering of novel and “*intriguing findings*”, we would like to clarify the following points. As explicitly stated, in the first part of manuscript, we validated the novel NoLMseq method using established knowledge, which naturally generated expected results. Thus, these results are not a limitation of this study. Instead, they demonstrated the validity of NoLMseq, so far, the first and unique genomic method to detect NADs in single nucleoli. To clearly separate the validation experiments from the exploratory application of NoLMseq, we reorganized the order of the figures and modified the text accordingly. The validation data of NoLMseq against established knowledge has been placed in **Figs. 1-3**, which also include some novel and intriguing data like the nucleolar proximity of ribosomal protein genes, followed by **Figs. 4-6** that uncovered novel and important results (i.e., genome remodelling upon ActD treatment, the mono-allelic features of NADs, the presence of two ESC populations with distinct NAD profiles and chromatin states, etc.). These are novel and important results that could only be obtained by a single cell technology like the NoLMseq and could not be captured by any other existing technologies.

Regarding the comment “*that a large fraction of NADs detach from nucleoli during the differentiation of ESCs into NPCs or upon nucleolar disruption has been reported previously by Bersaglieri et al. 2022 and Vertii et al. 2019*”, we would like to clarify two important points. The ESC to NPC differentiation previously analysed by Nucleolar-DamID in cell populations by Bersaglieri et al. was used to validate NoLMseq in another cell type than ESCs. We clarified this point in the revised manuscript. Moreover, this analysis added new information about the gene expression state upon detachment from nucleoli that was not reported previously. Second, our work is the first to demonstrate at both the genomic and single-cell levels that conditions disrupting nucleolar integrity such as the treatment with ActD causes NAD detachment from the nucleolus. Vertii et al did not provide genomic NAD data upon nucleolar disruption, an analysis that so far only NoLMseq could perform. Vertii et al. presented only three DNA-FISH results, one of which was not

statistically significant, from cells treated with hexanediol, without data showing that the nucleolus was affected/disrupted by this unusual condition to study nucleolus integrity.

Regarding the comment that “*The direct role of nucleolar association in gene regulation remains unproven*”, we would like to clarify that this was not the aim of this study. To prove this, it requires the use of other technologies, which we are currently establishing in our lab. Any unintended implication of a causal relationship has been corrected in this revised manuscript. However, we cannot disregard the data showing the high correlation between the derepression of genes at NADs upon disruption of nucleolar integrity and during NPC differentiation as well as the observation that biallelic NADs are in a more repressive state than monoallelic ones.

We respectfully disagree with the comment that “*the functional implications of these findings are not explored*” as the data of ESC to NPC differentiation and the disruption of nucleolar integrity have already provided functional significance (activated NAD-genes in NPC are linked to neural pathway and the ones activated upon treatment with ActD are implicated in p53 pathways). In this manuscript, we included scRNAseq data of ESCs showing that the two ESC populations not only differ in NAD profiles and chromatin features but also they might represent two distinct developmental states.

1. Lines 121-125: It is common for a single ES cell nucleus to harbour multiple nucleoli. Did the authors dissect only one nucleolus from each nucleus, or did they dissect all nucleoli from the same nucleus and combine them in one tube for later analysis? While this may not affect NAD identification, it could influence interpretations of NAD heterogeneity across cells. For instance, when the authors classify ESCs into two populations based on nucleolar contacts, could this be due to the capture of different nucleoli rather than true biological variability?

Author. We apologize to not have provided more details on the nucleolus structure of mouse ESCs. The large majority of mouse ESCs display one single nucleoli (see image below and also **new Fig. 1b**). During the microdissection, we only selected cells with one nucleolus cells (**Fig. 1b**). We clarified this point in the revised manuscript.

2. Lines 182-185: The authors found that the NAD/LAD (Type-2) regions exhibit overall lower CF values compared to NAD-only (Type-1) regions and conclude that “NAD/LAD domains rarely contact both nucleolus and NL in the same cell”. I find this logic unclear. It seems plausible that Type-2 regions represent a different type of heterochromatin than Type-1, and therefore associate with the nucleoli at a lower frequency.

Author. Our conclusions perfectly match with the ones of the Reviewer. The data in **Fig. 2e** simply showed that NAD-only regions are more frequently found in microdissected nucleoli than NAD/LAD regions. This makes sense, as NAD-only regions do not contact the nuclear lamina and therefore can be found more often in nucleoli, whereas NAD/LAD regions can also be located at the nuclear lamina. Because of this, we suggested that NAD/LAD might contact the nucleolus in one cell and the nuclear lamina in another cell. We modified the text to avoid confusion on this point. The important point here is that these measurements could only be achieved by single cell analyses. Moreover, these results further confirm that NoLMseq accurately identified NADs. We clarified this point in the manuscript.

3. The observation that ESCs can be classified into two populations, with chromosomes 12 and 19 preferentially associated with nucleoli in one group and chromosomes 16 and 18 in the other, is compelling. I suggest the authors conduct DNA-FISH to further validate this result.

Author. We have included additional computational data following the request of Reviewer 1 (point 5) that support the conclusion of our work. We performed bootstrap analysis with 1,000 random permutations of the data and confirmed that these two cell populations do not result from random fluctuations but they represent two populations with distinct chromatin features (**Extended Data Fig. 5a**).

To further validate the data and provide biological and functional significance of this finding, we performed scRNAseq of ESCs, as also suggested by the Reviewer in point 4. The ESC used in our study were cultured in the presence of serum. These conditions are well known to generate a certain heterogeneity. In particular, the expression of the pluripotency factor Nanog is heterogeneous in ESCs, giving rise to two major populations with distinct developmental potential: Nanog-high and Nanog-low. These two populations can be clearly visualized in the UMAP shown in **new Fig. 4e**. Notably, 4 out of 5 top genes with the highest nucleolus interactions and that were specific for population 2 show a similar expression profiles of Nanog-high cells and Sox2 positive ESCs while only 1 gene of population 1 show a similar profile (**new Fig. 4f, Extended Data Figs. 5b,c**). Notably, GO and pathway analyses showed that the specific NAD-genes of population 2 have significant enrichments in pathways linked to development and pluripotency whereas NAD-genes of population 1 are linked to metabolism and signalling pathways (**new Supplementary Table 2**). Thus, the two populations that were identified for a distinct NAD profile might represent two distinct developmental states of ESCs. The data together revealed structural heterogeneity in the ESC genome, resulting in two major populations distinguished by their genome organization around nucleoli, each characterized by distinct chromatin states and eventually different developmental states.

For the validation by DNA-FISH suggested by the Reviewer, we will require the simultaneous use of four distinct probes, each labelled with different colours. 2 probes for population 1 to demonstrate that these two NADs contact nucleoli in the same cells and 2 probes for population 2 to show that they contact nucleoli in the other fraction of cells. However, such a complex experiment is not feasible for us. Accordingly, all scLamin-DamID studies published so far have not shown validation by DNA-FISH. We believe the additional computational and experimental data we provided in this revised manuscript sufficiently validate the findings obtained through NoLMseq

4. Related to the last point, what are the functional implications of these distinct patterns of NAD organization in different ESCs? The authors should explore the relationship between NAD heterogeneity and gene expression by performing RNA-FISH. Additionally, the authors could analyze available scRNA-Seq data to determine whether gene expression from NADs in different ESCs can also be categorized into two populations.

Author. As described above, we performed the scRNAseq showing that the two populations might reflect two different developmental potentials.

RNA-FISH for nascent transcripts (targeting of introns) are usually performed for highly transcribed genes or reporter constructs whose signal can be easily detected. NAD genes are generally low expressed. This is also evident in the image below comparing Nanog and Sox2 expression with the other NAD genes of populations 1 and 2. We have looked very carefully in the literature and no RNA-FISH has been performed for genes at LADs. Therefore, we believe that RNA-FISH of NAD genes may not be feasible, as the low expression of NADs is unlikely to produce a detectable signal. Moreover, we think that addition of new computational (i.e., bootstrap analysis with 1,000 random permutations) and experimental data (i.e. scRNAseq) have clearly demonstrated the presence of these two ESC subpopulations and their potential functional significance.

5. Lines 226-228: The authors observe that genes within domains that detach from nucleoli upon ESC differentiation to NPCs (ESCsp-NADs) are more highly expressed in NPCs, leading them to conclude that detachment facilitates gene activation. However, the authors should include appropriate controls for the analysis in Figure 4f-i, such as genes within common NADs, NPCsp-NADs, and non-NADs within Compartment A, to support this conclusion. Ideally, RNA-FISH should also be performed to validate this finding.

Author. We have included this analysis in the revised manuscript (**new Extended data Fig. 4b**). The data confirmed the previous results by showing that ESC_{sp}-NADs are more expressed in NPCs. Genes at NPC_{sp}-NADs are very few (29) and showed a certain tendency to be more expressed in ESCs than in NPCs, although this difference was not significant. Genes at common NADs show only minor changes between ESCs and NPCs as well as non-NADs within A compartment.

6. While this study demonstrates that the ESCsp-NADs genes are activated in NPCs, a previous study from the same group (Bersaglieri et al. 2022) stated that the ESCsp-NADs genes do not exhibit significant gene expression changes and that the detachment from nucleoli unlocks these genes for activation in later stages. What is the reason for this discrepancy? The NoLMseq method likely revealed the real changes in NAD organization. However, the authors should discuss the differences between the two studies in more detail.

Author. This important point was introduced in the Discussion section (now lanes 374-376). “*The accuracy of NoLMseq in detecting NADs relative to cell population studies is also shown by the identification of ESC_{sp}-NADs that contain genes activated in NPCs, which could not be detected in previous cell population studies*”¹⁹. We extended this part by explaining the major difference between the two methods.

The NADs identified in ESCs and NPCs using NoLMseq provide a more quantitative assessment of nucleolar associations compared to our previous approach using Nucleolar-DamID (Bersaglieri et al., 2022). While nucleolar DamID allows to classify genomic regions broadly as associated or not associated with the nucleolus, and these data were validated by DNA-FISH and rDNA-HiC, it lacked the sensitivity to quantify the degree of association. In contrast, NoLMseq enables the quantification of nucleolar CF and the use of stringent criteria to robustly define NADs, as in the case of ESC_{sp}-NADs, defined as high-confidence NADs in ESCs that were consistently excluded from the nucleolus (>25% CF in ESCs, <10% CF in NPCs). Thus, this quantitative resolution provides a more nuanced view of nucleolar genome organization and enabled us to detect changes that were only partially captured with Nucleolar-DamID. We included this information in the Discussion of the revised manuscript.

7. In Figures 5h and 5i, the authors show that genes within NADs that detach upon ActD treatment are enriched in upregulated genes versus downregulated genes. While this finding is intriguing, I recommend that the authors compare the fraction of upregulated, downregulated, and unchanged genes across three groups: detached NADs, NADs that remain attached, and non-NAD regions. Statistical analyses, such as the Chi-square test, should be performed to help assess the significance of these observations.

8. Lines 277 to 279: The authors stated, “The detachment of genes from nucleoli, when nucleolar integrity is compromised, leads to their transcriptional activation”. I think this is an overstatement, as the authors have not conducted experiments to prove causality. Additionally, even within detached NADs, while 906 genes show upregulation, 705 genes are downregulated (Figure 5i). Thus, the effect of nucleolar association on gene expression appears to be facilitative and context-dependent. The authors should also exercise caution when making similar claims in the discussion.

Author. We revised the text to eliminate any unintended implication of a causal relationship.

We would like to thank the Reviewer for the insightful question, which prompted us to reanalyse the data using other criteria, which this time aim to determine how many genes at NADs are significantly regulated by ActD treatment, rather than how many regulated genes are located at NADs, as shown in the previous version. We believe this new analysis provides a more precise measurement and more directly addresses the question of how NAD-associated genes are regulated following ActD treatment. First, we analysed genes located in high confident NAD-only regions (CF > 30%) in ESCs and found that 17% of these NAD-genes (507) were upregulated upon ActD treatment whereas only 9% (260) were downregulated (**new Fig.5i**). In contrast, genes in non-NADs (CF < 20%) showed similar number of up- and downregulated genes upon ActD treatment. These results suggest that the loss of nucleolar integrity might play a significant role in the derepression of NAD-genes. Moreover, we also found that about half of the upregulated NAD-genes remain in contact with nucleoli in ESC+ActD and, accordingly, the majority of them (71%) were located at rDNA-chromosomes. These results suggest that the nucleolus may also lose its repressive properties under these nucleolar stress conditions, although further investigation is required to fully understand these properties, but this goes outside the aim of this work.

It is important to note that not all genes at NADs are expected to be reactivated within 24 hours of ActD treatment, as also key transcription factors may still be absent. Conversely, the downregulated genes may represent a subset that requires nucleolar association for their activity, which is consistent with our results showing the presence of some actively transcribed genes at the nucleolus. However, this investigation is out of the scope of this work and will be fully addressed in our future studies.

9. The finding that most NADs associate with nucleoli in a monoallelic manner is also intriguing. As with my previous comment (point 4), I suggest the authors either perform RNA-FISH or analyze available scRNA-Seq data to investigate how the allelic association with nucleoli affects gene expression.

Author. We share the Reviewer’s interest in the relationship between allelic nucleolar association and gene expression. We would like to strength that several data have supported the mono-allelicity of NADs like DNA-FISH (**Fig. 6d**) and the higher CF of biallelic NADs compared to monoallelic NADs (**Extended data Fig. 7d**). Furthermore, analyses of histone modifications and gene expression between mono- and bi-allelic NADs indicate that bi-allelic NADs are in a more repressed chromatin and gene expression states (**Figs. 6f,g**).

We followed the suggestion of the Reviewer and generated high-depth, allele-resolved scRNAseq data from the hybrid F123-ESCs (~30,658 UMIs per cell, compared to the current ~1,600 UMIs per cell in Bonora et al., 2021). Despite this increased depth, the resolution remains insufficient to reliably distinguish monoallelic expression, as we were only able to identify 26 genes with reliable allele-specific expression, defined as having greater than one read per allele per cell on average (this number was zero in publicly available F123 scRNAseq datasets). These genes are all highly expressed, do not localize to nucleoli, and exhibit the expected biallelic expression patterns across most cells. In contrast, genes NADs are generally lowly expressed, with average expression levels below one read per allele per cell. Consequently,

expression at the single-cell level was often binary (i.e., “0” or “1”), which complicates robust inference of allelic expression states. Although a monoallelic expression pattern can superficially appear in such sparse data, we believe this primarily reflects limitations of detection sensitivity rather than true regulatory bias. For these reasons, we believe that current scRNAseq approaches do not have the resolution necessary to reliably assess whether allelic association with nucleoli influences gene expression. Similarly, due to the generally low expression levels of NAD-associated genes, RNA-FISH is unlikely to yield interpretable allelic expression patterns for the majority of targets. Nevertheless, we share the Reviewer’s enthusiasm for this intriguing observation and believe that future developments in spatial transcriptomics or more sensitive allele-resolved methods may help address this question more definitively.

Single-cell dynamics of genome-nucleolus interactions captured by nucleolar laser microdissection (NoLMseq)

Corresponding Author: Professor Raffaella Santoro

Version 0:

Reviewer comments:

Reviewer #1

(Remarks to the Author)

The authors have further addressed my remaining points, but I think some more work is to be done for point 3, which I think provides a critical assessment of the data quality. The analysis that I suggest is highly feasible - the data are already there.

GENERAL:

The authors now list a few external code packages that they used for their analysis, but still do not make any of their own code available. A lot of data processing was done, and custom R and Python scripts are mentioned for specific analyses. All code should be publicly shared, so others can reproduce and check the results.

1. OK. I had missed that the images showed a cluster of cells, not a single cell. Figure 6d shows nuclei with 2 or 3 nucleoli, which further led me to think that multi-nucleolar cells are common.

2. OK. As mentioned above, please make code of these simulations available.

3. Thank you for the clarification. What the reader needs to understand is: which proportion of the captured DNA in each single-cell is truly NAD, and which proportion comes from chromosome chunks that happen to be below and above the nucleoli (and thus are not NADs, but "bycatch" DNA)? In our dialog we are gradually getting closer to this estimate, but I think we are not entirely there yet. A more precise calculation is warranted. Here is what I mean:

The new text now states that the nucleolus + 0.5 μ m NAD rim together occupy on average 65% of the excised cylinder. For the sake of my argument, let us assume that one-third of this volume consists of the rim, and two-thirds of the actual nucleolus. Thus, the rim (= NADs) occupies $65/3 = \sim 22\%$ of the cylinder volume. Note that the nucleolus itself contains barely any DNA. The remaining 35% (100%-65%) of the captured cylinder consists of nucleoplasm and cytoplasm. The nucleoplasm contains non-NAD DNA, so it is important to know which part of this 35% is actually nucleoplasm. Let us assume that this number is 20%. If all of these numbers are correct, then the captured DNA from a single cell was derived from 22% (NADs) + 20% (nucleoplasmic DNA) of the cylinder volume. Thus, assuming similar DNA concentrations in NADs and nucleoplasm, nearly half (20/42) of all sequenced DNA would be nucleoplasmic DNA, and a bit more than half (22/42) would be NADs.

Of course, I had to make a couple assumptions because the provided data was incomplete. But this rough calculation illustrates that as much as half of the captured DNA in each cell might NOT be from NADs.

Because the authors already have the obtained Imaris images, it should now be feasible to repeat this calculation with the real numbers, for each of the 6 cells (more cells would give a better view of cell-to-cell variability, which is so prominent in figure 1c). It is quite possible that the estimated proportion of non-NAD DNA turns out to be much lower than my back-of-the-envelope-calculation, which would take away my concern, and it would be great to share this information with the readers. But if the calculations indicate that a significant proportion of the data from each cell are non-LAD, then the readers need to know this too.

Finally, I fully agree that "pseudo-bulk" analysis of the 53 cells will largely average out the randomly captured nucleoplasmic chunks. But NoLM-seq is claimed to be a single-cell method (e.g. in the Title), so it is important to determine the single-cell

accuracy as carefully as possible.

4. The new figure 1h that is mentioned in the rebuttal seems to be missing in Figure 1.

9. "Proximity" would be more accurate, but I leave it up to the Editor to decide whether "Interactions" is OK in the Title.

10. The ACF analysis is now adequate. However, from the main text it is unclear why the authors favour the interpretation that the difference in decay is reflecting biological differences, rather than differences in noise levels. It seems to me that higher noise levels in NPCs cannot be ruled out. Also, the main text could explain how the fitted a and b should be interpreted.

11. What I meant is that it is not effective to only provide information in the rebuttal without adding it to the manuscript. Readers may have the same question.

=====Reviewed by Bas van Steensel, review request received: 08 August 2025; completed 13 August 2025. It is my standard policy to sign and date *all* of my manuscript review reports, regardless of my comments and recommendations. All correspondence about this manuscript should go via the editor. PLEASE DO NOT REMOVE THIS NOTE =====

Reviewer #3

(Remarks to the Author)

The authors have adequately addressed my previous questions in this revised manuscript. Overall, I find the study suitable for publication in Nature Communications.

Single-cell dynamics of genome-nucleolus interactions captured by nucleolar laser microdissection (NoLMseq)

Walavalkar *et al.*

Response to the Reviewers

We thank the Reviewers for the positive and constructive comments. In this revised manuscript, we have provided additional data addressing all the requests of Reviewers 1 and 3, included additional information in the method section, and modified the text to clarify points that were not sufficiently clear. The additional data strengthen the conclusions of our work.

List of changes

Modifications in the text were highlighted in red.

The new data are listed here below.

	Revised manuscript	Previous version	Description	Reviewer
1	Fig. 1b	Fig. 1b	New images of nucleolus microdissection	Rev. 1
2	Figs. 4f	---	Additional analyses of scRNAseq	Rev. 2
3	Ext. Data Fig. 1a	Rebuttal letter	Multiple images of nucleolus microdissection	Rev. 1
4	Ext. Data Fig. 1d	---	Imaris 3D IF images	Rev. 1
5	Ext. Data Fig. 1e	---	Cumulative histogram of genome-wide nucleolar CF values for randomized sequences for 53 nucleoli matching the genome coverage of each of the 53 ESC nucleoli measured by NoLMseq	Rev. 1
6	Ext. Data Fig. 3c-d	---	Calculation of R and P values of scatter plots	Rev. 1
7	Ext. Data Fig. 4a	---	Autocorrelation analysis of ESCs and NPCs with random samples	Rev. 1

Reviewer #1 (Remarks to the Author):

GENERAL:

The authors have addressed several of my concerns (see below; numbering as in the rebuttal). But my major points 1-4 need additional attention. Point 3 is a very difficult one: from the responses and new analyses, it seems to me that NoLM-seq has an intrinsic limitation that causes it to be quite noisy - it often captures chromosomal regions that are not nucleolus-associated, but happen to be above or below the nucleolus. This substantially dampens my enthusiasm for the technique. Perhaps the method will be only really useful when applied to very flat cells with "pancake" nuclei such as fibroblasts? At the very least, the authors should be very explicit about this limitation in the main text (instead of tucking it away in the Methods), so readers can decide for themselves. Points 2 and 4 are also relevant in this context.

Author. Below we provide our detailed clarifications. We believe that several of the points raised may stem from misunderstandings of the data and interpretations presented. We appreciate the reviewer's initial enthusiasm for NoLMseq and hope that our responses will help to clarify these aspects and restore confidence in the strength and novelty of our approach.

Important general remark: all code for data processing and data analysis (ideally a separate script or markdown file for each figure) should be made available, preferably on Github. Currently this is missing. Combined with the very cursory description of the computational analyses this is not acceptable. It seems misleading that the Reporting Summary states: "No software was used for data collection" - obviously a lot of computational work was needed for the processing and visualization of the data.

Author. We would like to clarify that the statement in the Reporting Summary "No software was used for data collection" refers specifically to the experimental phase. Thus, our statement is correct since

no automated or instrument-specific software was employed to collect the data (i.e., nucleoli microdissection) as clearly described in the method section.

All computational work was conducted during the analysis phase. No new software or pipelines were developed for this study; all analyses were performed using publicly available and well-established tools. All software and tools were cited in the Methods section of the manuscript and were used followed standard guidelines as per their documentation, all of which are available on their associated Github repositories. Most other graphs were generated using Graphpad Prism. In this revised version, we included additional details in the methods.

MAJOR POINTS:

1. The new figure 1b still shows only one nucleolus. This is not sufficient for giving the readers a good view of how the microdissections are done. Please add a set of ~ 10 representative images collected throughout this project.

Furthermore, the authors only address the issue of multiple nucleoli *in their rebuttal*, but do not discuss this *in the manuscript*: how often did the cells have multiple nucleoli for the collected data, and how was this issue handled (were all nucleoli, including very small ones, captured; or was only one big nucleolus captured?). Bottom line: precise documentation and transparency in the manuscript is needed.

Minor point: It is puzzling that the authors respond in their rebuttal: "It is certainly possible to cut two nucleoli, however, we did not include this in our analysis since the large majority of ESCs have one large nucleolus." Then why did they show in their rebuttal two images of nuclei in which multiple nucleoli are captured?

Author: This request was addressed in our previous response, showing a series of images of microdissection. We have now included these images in the manuscript (**new extended data Fig. 1a**). Moreover, we would like to clarify that the laser-microdissection of nucleoli has reliably worked throughout the study and has never posed any problems.

As clearly stated in the manuscript (Results, line 118; Discussion, line 438), the vast majority of ESCs contain a single large nucleolus, which makes this cell type an ideal model for NoLMseq. We did not analyse the few cells with multiple nucleoli and therefore did not discuss handling multiple nucleoli in the manuscript, as it is not relevant to the data presented. Moreover, we have clearly stated that NoLMseq is a method for cells with single nucleoli (Discussion, lines 453-458).

Minor point. The images in the rebuttal show multiple cells with one nucleolus each rather than one cell with multiple nucleoli, as the Reviewer's comment seems to imply. This point was perhaps misunderstood.

Finally, we kindly disagree with the comment of the reviewer that "*precise documentation and transparency in the manuscript is needed.*" Our manuscript has provided transparent and precise documentation of our methodology (i.e., 4 pages were dedicated to describing NoLMseq). Nevertheless, we included additional information in the Method to improve clarity.

2. I appreciate the newly added random simulations. But there seems to be a flaw in this: I understand that the 53 random datasets all were taken to have 15% coverage. The real data however show a broad distribution of coverages (figure 1f). The random datasets should have the same distribution of coverages. I could not find a description of the randomization in the Methods section, so it is not clear how this was exactly done.

Author: We performed the requested analysis with the random datasets with the same distribution of coverages (**new Fig. S1e**). As expected, there is no difference. We included additional details in the method section of this revised manuscript how this analysis was performed.

3. The new estimate that only 34% of the captured volume is nucleolar puts the results in a very different light. If two-thirds of the captured volume is above and below the nucleoli, and considering that nucleoli (one-third of the volume) contain very little DNA, then this method primarily captures non-nucleolar DNA! This reduces my enthusiasm for the method; most of the domains visible in figure 1c may be simply chunks of chromosomes that happen to be above or below the captured nucleolus. This explains why there seems so little concordance between individual nuclei, unlike Jop Kind's single-cell maps of lamina interactions.

Furthermore, proper documentation is needed of the 34%: the readers need a figure showing how the measurement was exactly done, plus a plot of the distribution of percentages across a set of ~50 nuclei.

Author: We think that there has been a misunderstanding.

The Reviewer requested the nucleolus volume of the microdissection. We provided this value (34%) using Imaris on immunofluorescence images stained with the nucleolar marker NPM1, while nuclear

volumes were defined using DAPI. Since the reason of this request was not clear to us, we did not discuss it in the results but included it in the method section, since as such it was not informative to explain the NoLMseq method. Indeed, NADs are located around the nucleolus rather than residing within the nucleolus. Now that we understand the rationale behind the reviewer request, we would like to clarify that **the relevant information for understanding what is captured by NoLMseq is the combined volume of the nucleolus and surrounding NADs, not the nucleolus alone**. As NADs were recently reported to form a ring around nucleoli marked by H3K9me2 in ESCs with a ~0.5-0.6 μm width, we used this spatial definition and estimated that nucleolus+NAD occupies ~65% of the volume in the laser microdissections.

An important clarification, already described in the manuscript, is that NAD identification does not rely on sequencing from a single microdissected nucleolus. Instead, it results from the integration of data from 53 individual microdissections. With the new information of the nucleolus+NAD volume in the microdissection, we can now provide additional quantitative measurements demonstrating the high confidence of NAD identification by NoLMseq. Specifically, when integrating across 53 microdissected cells using a 20% CF threshold, the measured ~15% NAD coverage per cell, and accounting for the nucleolus+NAD volume, **the probability of incorrectly classifying a non-NAD region as a NAD is just 0.000144%**. Moreover, since the mouse genome comprises of 27,904 100kb-bins, the number of false-positives in the data would be 0.04, thus less than a single bin! All of this updated analysis has been included in the results and method sections of the revised manuscript.

In any case, we believe that **the volume captured in the microdissection does not play a critical role in NAD identification**. Even when using the 35% nucleolus-only volume, the CF approach we employed shows that the probability of incorrectly classifying a non-NAD region as a NAD remains very low, at just 0.072% (i.e., 20 100kb-bins).

Finally, we believe the reviewer may have overlooked our stringent CF thresholding criteria for defining NADs, which were based on 53 microdissected nucleoli. As discussed in the manuscript, the predictable inclusion of genomic regions located below and above nucleoli, without applying any thresholding, would indeed result in a high number of false positives. However, this does not reflect our methodology to NAD definition

ESCs exhibit a highly homogeneous nuclear structure. Accordingly, our Imaris measurements of the nucleolus+NAD volume from six microdissected cells have a margin of error of only $\pm 5\%$. Given this low variability, there is no need to measure 50 cells. Moreover, as indicated above, the estimated volume has minimal impact on the CF-based method we applied to identify NADs using NoLMseq.

Jop Kind's maps of LADs are irrelevant to this study since it is biologically and methodologically a totally different system.

4. I respectfully disagree. Binary analyses always require arbitrary cutoffs, which can greatly affect the outcome. Visualization of continuous data is much more informative. For the sake of transparency, scatterplots should really be shown. Of course the authors may then discuss their interpretation of potential caveats, but handwaving in a rebuttal only is not acceptable. Furthermore, the new scatterplot in Extended Fig 3b (comparing the results to SeqFISH, another method) shows the value of scatterplots: a clear forked distribution is visible, pointing to a systematic bias in at least one of the two methods. It seems to me that systematic analysis of scatterplots comparing NoLMseq to DamID, SPRITE and SeqFISH could help to find out which method deviates from the others.

Moreover, I still do not understand how the overlap in figure 1h was calculated. Please provide a description including formula in the Methods. Considering point 3 above, how can the overlap be close to 100%? This really does not make sense.

Author: As described extensively in both our previous response and the Discussion section of the manuscript, Nucleolar-DamID is inherently a qualitative method, not a quantitative one. Therefore, directly comparing the non-quantitative NAD scores from Nucleolar DamID, which tend to be very similar across regions, with the quantitative CF values derived from single-nucleolus data would not be scientifically appropriate and could lead to misinterpretation.

Regarding SPRITE, the scores are not publicly available and reproducing them would require considerable expertise in the SPRITE data processing pipeline, which is beyond the scope of this study. For this reason, we initially opted for a broader genomic coverage comparison of common NAD regions (**previously shown in Fig. 1h**). However, we have now removed this panel, as it appeared to create more confusion than clarity.

Importantly, the core conclusions of our study remain well supported by robust and quantitative comparisons, particularly those between NoLMseq and seqFISH+ (**now presented in Fig. 1h and Extended Data Fig. 3b**). This represents an orthogonal and more meaningful comparison, as they both are quantitative single-cell technologies. Finally, the deviation of the scatter plot in **Extended Data Fig.**

3b occurs at CF >20%, which is the threshold used to define NADs. Additionally, the two datasets not only differ in methodology but also in resolution, NoLMseq at 100 kb and seqFISH+ at 1 Mb, explaining why the results are not identical.

5. OK, thanks for adding these new analyses → ESC population based on NADs (Fig. 4)

6. OK. → ESC and NPC (Fig. 3)

7. Thanks for the scatterplot, which boosts confidence in the ribosomal genes observation.

8. OK. → Bi-allelic and mono-allelic NADs (Fig. 6)

MINOR POINTS:

9. I still think that the word "interactions" in the title is inappropriate. It is odd that the authors agree with me that this word should be avoided, but that they keep it in the title. Proximity is better, particularly considering point 3 above.

Author. We appreciate the reviewer's continued feedback on this point. While we understand the concern regarding the term "*interactions*", we respectfully maintain our choice to use it in the title for the following reasons. As indicated in our previous response (quoted below), we have deliberately avoided using the term "*interactions*" throughout the main text of the manuscript to avoid any potential overstatement. However, we believe that in the context of the title, "*interactions*" remains the clearest and most accessible term for a broad readership. This usage aligns with established conventions in the field, for example, numerous studies on LADs have used "*interactions*" in their titles to convey nuclear spatial associations, even when referring to proximity-based measurements.

"We agree. Accordingly, the term 'interactions' was not used in the rest of the manuscript. However, we think that for a title the term 'interactions' would be much clearer for a broad readership, as has been done in the past for many LADs papers."

We hope the reviewer will understand our rationale and accept this terminology in the interest of clarity and consistency with prior literature.

10. I disagree that the ACFs are similar. Clearly, the autocorrelation is worse in NPCs (shifted substantially to the left!), and hence the data are probably more noisy. I do not understand how the Pearson correlation was calculated, but it seems not the right measure to determine the shift in ACF.

Author. We agree with the reviewer that the ACF curves for NPCs are shifted to the left, indicating reduced autocorrelation. However, as described in the manuscript, we believe this reflects underlying biological differences between the two cell types (i.e., NPC has less NADs than ESCs). As described in details in our previous response to this point, these data are consistent with previous work "*the reduction of contacts with nucleoli in NPCs compared to ESCs is consistent with previous work using Nucleolar-DamID and rDNA-HiC (PMID: 35304483)*".

To further assess whether this shift could be attributed to technical noise, we generated random datasets for both ESCs and NPCs, matched for genome coverage per sample. As expected, these random datasets showed substantially lower autocorrelation values closer to zero, supporting the interpretation that the observed ACF patterns in actual nucleolar samples are biologically meaningful rather than artifacts. The random NPC datasets also displayed a slight leftward shift compared to random ESC datasets, likely due to lower overall NAD coverage in NPCs compared to ESCs. We have removed the Pearson correlation analysis from the manuscript and describe values only for the autocorrelation and decay constants. We modified the text to clarify these points.

11. (i.e., success rate of NoLMseq) OK, but please document this in the manuscript, ideally with small table or figure, not just in a response in the rebuttal.

Author. We respectfully disagree with the need to include this information in the main manuscript. The relevant details have already been provided in our rebuttal, which will be publicly available and offers sufficient context for readers interested in the technical aspects. Our aim is to present a scientific method, not to market a product. The robustness and utility of our approach are clearly demonstrated through data quality, reproducibility, and comparative analyses with established methods. Including a table of "success rates" would not add meaningful scientific value.

12. OK. → "bimodal" term

13. The new scatterplots shed new light on the links with H3K27ac and H3K9me2. The correlations turn out to be extremely weak! Correlation coefficients are not provided, but by eye they seem <0.2 . This should be discussed explicitly, and I think it is somewhat misleading to have these scatterplots tucked away in the Extended data.

Author. First, we would like to clarify that all relevant data have been presented transparently, and we respectfully disagree with the statement that any data were tucked. As explicitly stated in our previous response “*we decided to place them (i.e., scatter plots) in the Extended data Figs. 3c-e since they are less informative than the boxplots showed in Figs. 2f-h. Indeed, the boxplot representation of the data allows comparisons with genomic domains within the A compartment, the NAD-only, and NAD/LAD regions.*” Moreover, the box plots demonstrate that NAD-only regions with CF $>50\%$ have significantly lower H3K27ac and higher H3K9me2 levels compared to regions with lower CF, an insight that the scatter plots cannot convey as effectively. Therefore, we believe that the box plots provide a more informative visualization and are appropriately placed in the main figure.

We have now included the requested r and P values in the scatter plots shown in Extended Data Figs. 3c-e. As expected, the R values are modest; however, the decrease in H3K27ac levels and the increase in H3K9me2 content relative to nucleolar contact frequency are statistically significant and align well with the box plot representations in Figs. 2f-h, as well as the corresponding descriptions in the Results section. We described these results in the manuscript.

14. OK. → legend correction, Fig. 3b

15. OK. → NAD and LAD in ESC vs NPC

16. OK. → NAD-only and NAD/LAD, Fig. 2e

=====
Reviewed by Bas van Steensel, review task received: 12 May 2025; completed 24 May 2025. It is my standard policy to sign and date *all* of my manuscript review reports, regardless of my comments and recommendations. All correspondence about this manuscript should go via the editor. PLEASE DO NOT REMOVE THIS NOTE =====

Reviewer #2 (Remarks to the Author):

The revised manuscript has addressed the reviewer's comments adequately. It is suitable for publication.

Author. We thank the Reviewer for the appreciation of our study.

Reviewer #3 (Remarks to the Author):

In this revised manuscript, Walavalkar et al. have incorporated new experiments and data analyses that clarify the physiological implications of the NAD organization revealed by NoLMseq. In particular, by performing scRNA-Seq in ESCs, the authors show that distinct nucleolar-associated domain (NAD) organization patterns in two ESC populations may correlate with the expression of genes within different NADs. Moreover, the authors performed additional analyses to further support the conclusion that dissociation of NADs from the nucleolus during ESC-to-NPC differentiation, as well as upon Actinomycin D treatment, is associated with derepression of genes. The authors have also made textual edits that improve clarity and corrected some overstatements.

Overall, I find that these revisions have significantly strengthened the manuscript. Most of my earlier concerns have been addressed; however, I do have a few remaining points that require further attention by the authors.

Author. We thank the Reviewer for the appreciation of our study. Below you will find the description of the changes according to Reviewer's request.

Major points:

1. I appreciate the authors' efforts to integrate scRNA-Seq with NAD mapping to illustrate that distinct nucleolar contact patterns in ESCs at the single-cell level may relate to differential gene expression and cellular states. By examining the expression of genes within NADs specific to the two ESC populations, the authors show that genes in population 1 NADs tend to co-express with pluripotency markers Nanog and Sox2, while those in population 2 NADs do not. This is an interesting observation. However, the current analysis is based on only five genes per NAD group. I suggest that the authors perform more comprehensive statistical analyses across a broader set of genes with detectable expression in the scRNA-Seq dataset to rigorously assess whether genes within the same NAD group show higher expression correlation compared to genes from different NAD groups.

Author. We thank the Reviewer for this insightful suggestion. We expanded our analysis to include the top 30 genes, selected from a total of 282 genes of population 1 and 126 genes of population 2, that showed the highest nucleolar interactions and were specific to either population 1 or population 2 (now shown in the **new Fig. 3f**). This broader analysis confirmed our initial observation: a substantial fraction of population 2 NAD genes (22) co-express with pluripotency markers such as Nanog and Sox2 in clusters 0-3, whereas only few NAD genes of population 1 (12) exhibit this pattern with 13 genes mainly expressed in cluster 7. These results are consistent with our previous findings based on UMAP and dot plots of the top 5 nucleoli-interacting genes.

Minor points:

2. Regarding lines 187 and 193 and my previous point #2: While I agree with the authors that it is plausible NAD/LAD regions may interact with the nucleolus in one cell and the nuclear lamina (NL) in another, the low CF values alone do not provide direct evidence for or against simultaneous nucleolar and NL contacts within the same nucleus. For instance, one could imagine a scenario where a NAD/LAD region contacts the nucleolus in 15% of cells, and in all such cases, it also contacts the NL. To avoid overinterpretation, I recommend removing the sentence in lines 186–188: "Since this classification of NADs was obtained from ESC population, it has remained elusive whether NAD/LAD regions contact both nucleoli as well as the NL in the same or different cells."

Author. We removed the sentence

3. In line 224, the authors show that by combining the more quantitative NoLMseq data and more stringent criteria, they were able to more accurately define the ESCsp-NADs and reveal the correlation between NAD changes and gene derepression. I suggest the authors include a panel in the extended data figure to illustrate the differences between ESCsp-NADs identified by NoLMseq and those previously defined using nucleolar DamID.

Author. To further clarify the improved resolution of NoLMseq to detect NADs relative to the previous cell population based Nucleolar-DamID, we included new data showing the non-significant upregulation of genes located at ESC_{sp}-NADs identified by Nucleolar-DamID between ESCs and NPCs (**new extended data Fig. 4c**).

4. Lines 225–227: It would be helpful to mention here that genes located in NPCsp-NADs, common NADs, or in the A compartment do not exhibit significant expression changes.

Author. As suggested, we included a more detailed description of the **Extended Data Fig. 4b**.

5. Line 278: The authors suggest that ESC populations with distinct nucleolar contact patterns may represent “different developmental potentials.” However, these populations could simply reflect dynamic, fluctuating states of ESCs under in vitro culture conditions. I recommend softening this statement to avoid overinterpretation.

Author. We would like to clarify that our statement is grounded in extensive literature describing the distinction between ESCs cultured in 2i conditions and those maintained in serum/LIF. The latter are widely recognized as representing a more heterogeneous and developmentally advanced or “primed” pluripotent state. The terminology we used, referring to different developmental potentials, accurately reflects these established differences. Therefore, we believe the current wording appropriately reflects the existing knowledge without overinterpretation.

Single-cell dynamics of genome-nucleolus interactions captured by nucleolar laser microdissection (NoLMseq)

Walavalkar et al

Third response to Reviewers

We would like to thank Reviewer #3 for the positive and constructive comments and for supporting the publication of our work.

Here below, our response to the additional request of Reviewer 1. Changes in the text are highlighted in red.

Reviewer #1 (Remarks to the Author):

The authors have further addressed my remaining points, but I think some more work is to be done for point 3, which I think provides a critical assessment of the data quality. The analysis that I suggest is highly feasible - the data are already there.

GENERAL:

The authors now list a few external code packages that they used for their analysis, but still do not make any of their own code available. A lot of data processing was done, and custom R and Python scripts are mentioned for specific analyses. All code should be publicly shared, so others can reproduce and check the results.

Author. To clarify, our work did not develop any new software. At request of the Reviewer, we have now included the very simple command-line instructions used in our work on GitHub.

1. OK. I had missed that the images showed a cluster of cells, not a single cell. Figure 6d shows nuclei with 2 or 3 nucleoli, which further led me to think that multi-nucleolar cells are common.

Author. We find it difficult to accept this justification, as this mistake could have been avoided with a more careful evaluation of our work. We repeatedly documented in the original manuscript, the first revised version, and the initial rebuttal letter that the vast majority of ESCs contain a single large nucleolus. Unfortunately, this misinterpretation was taken as evidence to suggest that our microdissections were flawed, which had an unjustifiable negative impact on the evaluation of our work at NSMB.

2. OK. As mentioned above, please make code of these simulations available.

Author. These simple command-line instructions are now deposited on GitHub

3. Thank you for the clarification. What the reader needs to understand is: which proportion of the captured DNA in each single-cell is truly NAD, and which proportion comes from chromosome chunks that happen to be below and above the nucleoli (and thus are not NADs, but "bycatch" DNA)? In our dialog we are gradually getting closer to this estimate, but I think we are not entirely there yet. A more precise calculation is warranted. Here is what I mean:

The new text now states that the nucleolus + 0.5 μ m NAD rim together occupy on average 65% of the excised cylinder. For the sake of my argument, let us assume that one-third of this volume consists of the rim, and two-thirds of the actual nucleolus. Thus, the rim (= NADs) occupies $65/3 = \sim 22\%$ of the cylinder volume. Note that the nucleolus itself contains barely any DNA. The remaining 35% (100%-65%) of the captured cylinder consists of nucleoplasm and cytoplasm. The nucleoplasm contains non-NAD DNA, so it is important to know which part of this 35% is actually nucleoplasm. Let us assume that this number is 20%. If all of these numbers are correct, then the captured DNA from a single cell was derived from 22% (NADs) + 20% (nucleoplasmic DNA) of the cylinder volume. Thus, assuming similar DNA concentrations in NADs and nucleoplasm, nearly half (20/42) of all sequenced DNA would be nucleoplasmic DNA, and a bit more than half (22/42) would be NADs.

Of course, I had to make a couple assumptions because the provided data was incomplete. But this rough calculation illustrates that as much as half of the captured DNA in each cell might NOT be from NADs.

Because the authors already have the obtained Imaris images, it should now be feasible to repeat this calculation with the real numbers, for each of the 6 cells (more cells would give a better view of cell-to-cell variability, which is so prominent in figure 1c). It is quite possible that the estimated proportion of non-NAD DNA turns out to be much lower than my back-of-the-envelope-calculation, which would take away my concern, and it would be great to share this information with the readers. But if the calculations

indicate that a significant proportion of the data from each cell are non-LAD, then the readers need to know this too.

Finally, I fully agree that "pseudo-bulk" analysis of the 53 cells will largely average out the randomly captured nucleoplasmic chunks. But NoLM-seq is claimed to be a single-cell method (e.g. in the Title), so it is important to determine the single-cell accuracy as carefully as possible.

Author.

We would like to clarify what the readers will understand and what the Reviewer needs to understand.

As clearly described in the manuscript, readers will understand that the microdissections will inevitably include non-NAD sequences located above and below nucleoli, and to exclude them we applied a threshold in nucleolar contact frequency (20%) across the 53 microdissected nucleoli, which was then used to identify NADs in individual cells. They will also appreciate that the NADs obtained with this approach were validated using multiple orthogonal methods, which this Reviewer has previously acknowledged and accepted.

As explained in our previous response, the Reviewer needs to understand that the proportions of nucleolus volume (35%), nucleolus+NAD volume (65%), or, as now requested, NAD vs. non-NAD proportions within each microdissected cylinder are not critical for NAD identification by NoLMseq. It is the nucleolar CF threshold that distinguishes NADs from non-NADs in each single microdissection. While the Reviewer has referred to this as a "pseudo-bulk" approach, we would like to stress that this is not a shortcoming of NoLMseq but rather a fundamental aspect of single-cell analyses. Indeed, virtually all single-cell methods rely on some form of "pseudo-bulk" treatment, such as thresholds, aggregation, or modelling, to extract accurate biological signals from inherently sparse and noisy data.

We find that the assumptions suggested by the Reviewer for his new request to calculate NAD vs. non-NAD proportions within the microdissected cylinder to be flawed and likely to produce misleading results (see below).

- **Cylinder definition:** As clearly described in the Figure, Figure legend and Methods, the volume of microdissected cylinder used in our measurements includes only nucleoplasm and nucleolus, not cytoplasm.
- **DNA distribution:** The assumption that DNA is homogeneously distributed in the nucleoplasm is conceptually incorrect. Chromatin is more condensed around nucleoli than in the nucleoplasm. Since DNA concentration is not uniform and is higher around nucleoli, calculating NAD and non-NAD proportions under this assumption is conceptually incorrect.
- **NADs in nucleoli:** The assumption that there are no NADs within nucleoli is also inaccurate. While it is true that nucleoli inside contain less DNA overall, some NADs can be found within nucleoli or at the inner outer layer of the nucleolus (granular component).

For the sake of curiosity, we performed this calculation using the simplified assumptions of the Reviewer and obtained a 50% NAD proportion within the microdissected cylinder. However, this value would likely be higher, given the higher DNA concentration around nucleoli relative to the nucleoplasm. Thus, we will not include this value in the manuscript, as it is based on incorrect assumptions, would be scientifically misleading, and does not accurately reflect NAD biology.

We will not modify the current description of this section of the manuscript, as it accurately reflects the data and does not require further changes. Readers will understand.

4. The new figure 1h that is mentioned in the rebuttal seems to be missing in Figure 1.

Author. There is no new Fig. 1h.

9. "Proximity" would be more accurate, but I leave it up to the Editor to decide whether "Interactions" is OK in the Title.

Author. This point was extensively explained in all our previous responses.

10. The ACF analysis is now adequate. However, from the main text it is unclear why the authors favour the interpretation that the difference in decay is reflecting biological differences, rather than differences in noise levels. It seems to me that higher noise levels in NPCs cannot be ruled out. Also, the main text could explain how the fitted a and b should be interpreted.

Author. This point, including the noise, was extensively explained in all our previous responses and in the manuscript and the data are consistent with published work. The a and b values were defined in the M&M section. We have now included their definition in main text, as requested.

11. What I meant is that it is not effective to only provide information in the rebuttal without adding it to the manuscript. Readers may have the same question.

Author. This point (success rate) was extensively explained in all our previous responses.

As it happens for all experiments, the success rate mainly depends on the expertise of the researcher. Thus, including a table of "success rates" would not add meaningful scientific value. The robustness and utility of our approach are clearly demonstrated through data quality, reproducibility, and comparative analyses with established methods.

Readers interested in further details can either contact us directly or consult our rebuttal letters, which will be publicly accessible.

=====Reviewed by Bas van Steensel, review request received: 08 August 2025; completed 13 August 2025. It is my standard policy to sign and date *all* of my manuscript review reports, regardless of my comments and recommendations. All correspondence about this manuscript should go via the editor. PLEASE DO NOT REMOVE THIS NOTE =====

Reviewer #3 (Remarks to the Author):

The authors have adequately addressed my previous questions in this revised manuscript. Overall, I find the study suitable for publication in Nature Communications.

Author. We would like to thank Reviewer #3 for the positive and constructive comments and its support for publication of our work.